# Two new *Nothophytophthora* species from streams in Ireland and Northern Ireland: *Nothophytophthora irlandica* and *N. lirii* sp. nov.

Richard O'Hanlon[1,2]*, Maria Destefanis[1], Ivan Milenković[3], Michal Tomšovský[3], Josef Janoušek[3], Stanley E. Bellgard[4], Bevan S. Weir[5], Tomáš Kudláček[3], Marilia Horta Jung[3,6], Thomas Jung[3,6]

1 Department of Agriculture, Food and the Marine, Dublin, Ireland, 2 Queens University Belfast, Northern Ireland, United Kingdom, 3 Faculty of Forestry and Wood Technology, Mendel University in Brno, Phytophthora Research Centre, Brno, Czech Republic, 4 Massey University, School of Fundamental Sciences, Palmerston North, New Zealand, 5 Landcare Research, Auckland, New Zealand, 6 Phytophthora Research and Consultancy, Nußdorf, Germany

* r.ohanlon@qub.ac.uk

**Data Availability Statement:** All relevant data are within the paper.

## Abstract

Slow growing oomycete isolates with morphological resemblance to *Phytophthora* were obtained from forest streams during routine monitoring for the EU quarantine forest pathogen *Phytophthora ramorum* in Ireland and Northern Ireland. Internal Transcribed Spacer (ITS) sequence analysis indicated that they belonged to two previously unknown species of *Nothophytophthora*, a recently erected sister genus of *Phytophthora*. Morphological and temperature-growth studies were carried out to characterise both new species. In addition, Bayesian and Maximum-Likelihood analyses of nuclear 5-loci and mitochondrial 3-loci datasets were performed to resolve the phylogenetic positions of the two new species. Both species were sterile, formed chlamydospores and partially caducous nonpapillate sporangia, and showed slower growth than any of the six known *Nothophytophthora* species. In all phylogenetic analyses both species formed distinct, strongly supported clades, closely related to *N. chlamydospora* and *N. valdiviana* from Chile. Based on their unique combination of morphological and physiological characters and their distinct phylogenetic positions the two new species are described as *Nothophytophthora irlandica* sp. nov. and *N. lirii* sp. nov. Their potential lifestyle and geographic origin are discussed.

## Introduction

*Nothophytophthora* is a monophyletic sister genus of *Phytophthora*, and was erected in 2017 to accommodate several slow growing previously unknown oomycete species recovered from surveys of rhizosphere soil and streams in forest habitats in Europe, Asia and South America [1–3]. The main features differentiating *Nothophytophthora* from other closely related oomycete genera are the presence of a conspicuous, opaque plug inside the sporangiophore close to

**Funding:** ROH was funded by Department of Agriculture, Food and the Marine Ireland through the PHYTOFOR project, and by the Department of Agriculture, Environment, and Rural Affairs Northern Ireland. ROH also acknowledges a Royal Irish Academy Charlemont scholar award for funding part of this work. The authors TJ, MHJ, IM, MT, JJ and TK are grateful to the European Regional Development Fund for cofinancing the Project Phytophthora Research Centre Reg. No. CZ.02.1.01/0.0/0.0/15_003/0000453.

**Competing interests:** The authors have declared that no competing interests exist.

the base of most mature sporangia in all known *Nothophytophthora* species and intraspecific co-occurrence of caducity and non-papillate sporangia with internal nested and extended proliferation in several *Nothophytophthora* species. Jung et al. [1] described six species within the genus *Nothophytophthora*. Isolates of other potentially novel *Nothophytophthora* taxa have been isolated by several research groups during the last decade [4–7].

Ireland has a heavily modified landscape, with over 60% of the land cover devoted to agricultural grassland [8]. The natural vegetation of the island of Ireland would consist mostly of temperate deciduous forests [9], although at present just 11% of the land area is forested [10]. Consequently, the majority of research in plant pathology in Ireland has been focussed on agricultural pathogens. The diversity of oomycetes in natural and semi-natural habitats on the island of Ireland, comprising the Republic of Ireland and the UK country Northern Ireland, has not been well studied. O'Hanlon et al. [11] presented evidence for the presence of 27 species of *Phytophthora*, and speculated that at least a further 11 species probably remained to be found based on species records from the UK. Surveys of forests, horticultural premises, and public horticultural gardens in the past five years have produced first records of eight *Phytophthora* species and several other oomycete species previously unrecorded in Ireland [12,13].

In recent surveys of Irish and Northern Irish habitats for the EU regulated forest pathogen *Phytophthora ramorum*, collection and testing of *Rhododendron* leaves from wild plants and from water baits in streams revealed several oomycete isolates which morphologically resembled *Phytophthora* [12–14]. Preliminary ITS sequence analysis indicated that these slow growing isolates belonged to two previously unknown species of *Nothophytophthora*. In this study, morphological and physiological characteristics were used in combination with multigene phylogenetic analyses to characterise the two new *Nothophytophthora* taxa, compare them with the known species of *Nothophytophthora*, and officially describe them as *Nothophytophthora irlandica* sp. nov. and *Nothophytophthora lirii* sp. nov.

## Material and methods

### Ethics statement

This study was performed within the frame of the annual surveys of Irish and Northern Irish habitats for the EU quarantine forest pathogen *P. ramorum*. The surveys were funded by, and had oversight from, the National Plant Protection Organisations of both jurisdictions. No specific permissions were required. Our field sampling did not involve endangered or protected species.

### Isolate collection and maintenance

Baiting was performed in two and one streams in Ireland and Northern Ireland, respectively, (Fig 1) using young leaves of *Rhododendron ponticum* or *Rhododendron caucasicum × ponticum* 'Cunningham's White' as baits in mesh sacs floating on the water [12,13]. The baiting in the Ow stream in Ireland took place between early 2014 and late 2015, with a total of 10 baits being tested during that period. The baiting in the Shimna stream in Northern Ireland took place between mid-2017 and early 2018, with a total of 7 baits being tested. In addition, attached leaves with lesions of plants of *R. ponticum* near the Owenashad and Shimna streams were collected on each occasion and tested. Furthermore, naturally fallen necrotic leaves of *R. ponticum* and other hosts (e.g. *Fagus sylvatica*, *Fraxinus excelsior*, *Quercus petraea*, *Corlyus avellana*) floating in two streams in Ireland and one stream in Northern Ireland were also sampled [3]. Collections in the Owenashad stream in Ireland were conducted in March and December 2014; in August 2015; in July 2017; and in March, June, July and August 2018. Collections in the Shimna stream in Northern Ireland happened in February, April, June, July,

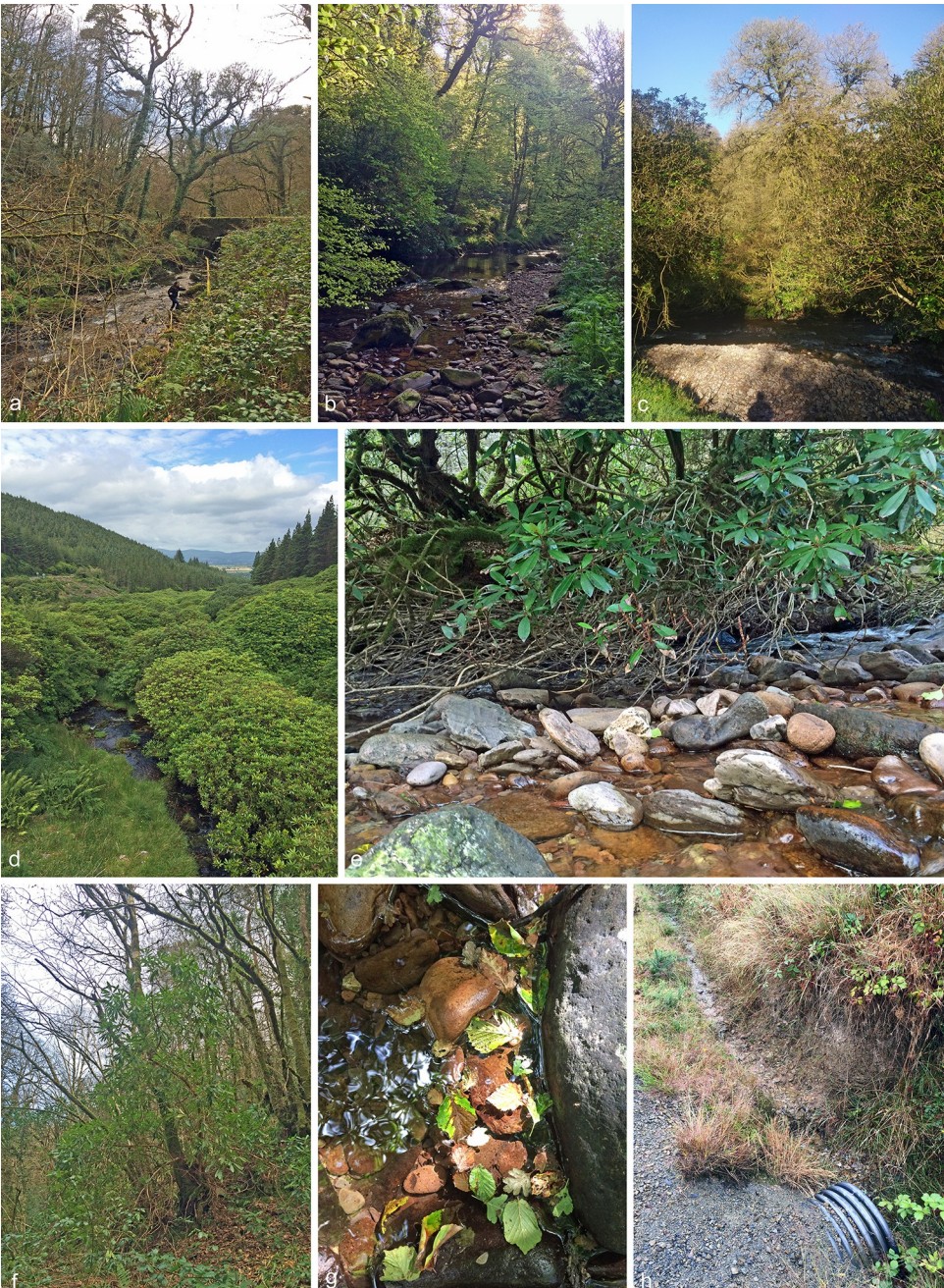

**Fig 1. Forest streams in Ireland from which *Nothophytophthora* spp. were isolated.** a–c. Owenashad River in a temperate mixed coniferous and deciduous forest in County Waterford; d–g. Ow River in County Wickford; d. riparian gallery of *Rhododendron ponticum* and planted conifer forest on the slopes; e. riparian *R. ponticum*; f. riparian mixed stand of *R. ponticum* and broadleaved trees; g. naturally fallen, partly necrotic leaves floating in the Ow River; h. small ditch which flows as tributary into the Ow River.

August and October 2017. In July 2015 a single collection of floating detached *R. ponticum* leaves was made from the Ara stream in Ireland. As all of the sampling described above was originally for the purpose of detecting a regulated organism (i.e. *P. ramorum*), no effort was made to record the number of leaves tested. However, lesions from several hundred leaves were plated during this research.

Isolations from necrotic areas of baiting leaves and naturally fallen leaves were performed using selective P5ARP agar (cornmeal agar with antibiotics; [15]). For all isolates, single hyphal tip cultures were produced under the stereomicroscope from the margins of fresh cultures on V8-juice agar (V8A; 16 g agar, 3 g CaCO₃, 100 ml Campbell's V8 juice, 900 ml distilled water). Stock cultures were maintained on grated carrot agar (CA; 16 g agar, 3 g CaCO₃, 200 g grated carrots, 1000 ml distilled water; [16,17]) at 10°C in the dark. All isolates of the two new *Notho-phytophthora* spp. are preserved in the culture collections maintained at the Agri-Food and Biosciences Institute, Belfast, Northern Ireland, and the culture collection maintained at Mendel University in Brno, Czech Republic. Ex-type and additional cultures were deposited in the CBS culture collection at the Westerdijk Fungal Biodiversity Institute (previously Centraalbureau voor Schimmelcultures CBS; Utrecht, The Netherlands). Details of all isolates used in the phylogenetic, morphological and temperature-growth studies are given in Table 1.

GenBank numbers for sequences obtained in the present study are printed in italics; ex-type isolates are printed in bold-type; t.s., this study;–, not available.

### DNA isolation, amplification and sequencing

For all *Nothophytophthora* isolates obtained in this study and for two isolates each of the six described *Nothophytophthora* species the Phire Plant Direct Master Mix (Thermo Fisher Scientific Inc., Gloucester, UK) was applied for direct PCR from fresh pure cultures growing on V8A, following the manufacturer's instructions. The mycelium extract diluted in dilution buffer was stored at –20°C. For *N. irlandica* and *N. lirii* five nuclear and three mitochondrial loci were amplified and sequenced. The internal transcribed spacer region (ITS1–5.8S–ITS2) of the ribosomal RNA gene (ITS) and the 5' terminal domain of the large subunit (LSU) of the nuclear ribosomal RNA gene (nrDNA) were amplified separately using the primer–pairs ITS1/ITS4 [18] and LR0R/LR6–O [19,20]. Partial heat shock protein 90 (*hsp90*) gene was amplified with the primers HSP90F1int and HSP90R1 as described previously [21]. Segments of the β-tubulin (*btub*), the mitochondrial genes cytochrome c oxidase subunit 1 (*cox1*), and NADH dehydrogenase subunit 1 (*nadh1*) genes were amplified with primers TUBUF2 and TUBUR1, COXF4N and COXR4N, FM84 and FM85, and NADHF1 and NADHR1, respectively, using the PCR reaction mixture and cycling conditions as described earlier [22,23]. Partial *rps10* gene was amplified according to the protocol provided by OomyceteDB (http://oomycetedb.cgrb.oregonstate.edu/protocols.html) using primer pair rps10_DB-FOR and rps10_DB-REV. Partial *tigA* gene amplification was performed using primers Tig_FY and G3PDH_rev according to Blair et al. [21]. For the six described *Nothophytophthora* species only *rps10* and *tigA* were amplified. All amplicons were purified and sequenced in both directions by Eurofins Genomics GmbH (Cologne and Ebersberg, Germany) using the primers of the PCR reactions except for the *tigA* amplicons for which primers Tig_rev and G3PDH_for were used [21]. Electropherograms were quality checked and forward and reverse reads were compiled using Geneious Prime v. 2021.0.3 (Biomatters Ltd., Auckland, New Zealand). Clearly visible pronounced double peaks were considered as heterozygous positions and labelled according to the IUPAC coding system. All sequences derived in this study were deposited in GenBank and accession numbers are given in Table 1.

### Phylogenetic analysis

The sequences obtained in this work for *N. irlandica*, *N. lirii* and the six described *Nothophytophthora* species were complemented with sequences of the latter retrieved from GenBank [1]. The sequences of the loci used in the analyses were aligned using the online version of MAFFT v. 7 [24] by the E-INS-I strategy (ITS) or the G-INS-I strategy (all other loci).

**Table 1. Details of *Nothophytophthora* and *Phytophthora* isolates included in the phylogenetic, morphological and growth-temperature studies.**

| Species | Isolate numbers [a] International and/or local collections | Origin Host / source | Location; year; collector; reference | GenBank accession numbers | | | | | | | |
|---|---|---|---|---|---|---|---|---|---|---|---|
| | | | | ITS | LSU | btub | hsp90 | tigA | cox1 | nadh1 | rps10 |
| **N. amphigynosa** [b] | CBS 142348; BD268 | Stream baiting | Portugal, 2015; TJ; Jung et al. 2017a [1] | KY788382 | KY788428 | KY788515 | KY788555 | MW427922 | KY788473 | KY788596 | MW427949 |
| *N. amphigynosa* [b] | CBS 142349; BD741 | Stream baiting | Portugal, 2015; TJ; Jung et al. 2017a [1] | KY788384 | KY788432 | KY788517 | KY788557 | MW427921 | KY788475 | KY788598 | MW427948 |
| *N. caduca* [b] | CBS 142350; CL328 | Stream baiting | Chile; 2014; TJ; Jung et al. 2017a [1] | KY788401 | KY788470 | KY788531 | KY788571 | MW427924 | KY788489 | KY788612 | MW427951 |
| *N. caduca* [b] | CBS 142351; CL333 | Stream baiting | Chile; 2014; TJ; Jung et al. 2017a [1] | KY788402 | KY788471 | KY788532 | KY788572 | MW427923 | KY788490 | KY788613 | MW427950 |
| **N. chlamydospora** [b] | CBS 142353; CL316 | Stream baiting | Chile; 2014; TJ; Jung et al. 2017a [1] | KY788405 | KY788450 | KY788535 | KY788575 | MW427926 | KY788493 | KY788616 | MW427953 |
| *N. chlamydospora* [b] | CBS 142352; CL195 | Stream baiting | Chile; 2014; TJ; Jung et al. 2017a [1] | KY788404 | KY788449 | KY788534 | KY788574 | MW427925 | KY788492 | KY788615 | MW427952 |
| **N. intricata** [b] | CBS 142354; RK113-1s | *Aesculus hippocastanum* | Germany; 2011; TJ; Jung et al. 2017a [1] | KY788413 | KY788440 | KY788543 | KY788583 | MW427928 | KY788501 | KY788624 | MW427955 |
| *N. intricata* [b] | CBS 142355; RK113-1sH | *A. hippocastanum* | Germany; 2011; TJ; Jung et al. 2017a [1] | KY788412 | KY788439 | KY788542 | KY788582 | MW427927 | KY788500 | KY788623 | MW427954 |
| **N. valdiviana** [b] | CBS 142357; CL331 | Stream baiting | Chile; 2014; TJ; Jung et al. 2017a [1] | KY788417 | KY788457 | KY788547 | KY788587 | MW427930 | KY788505 | KY788628 | MW427957 |
| *N. valdiviana* [b] | CBS 142356; CL242 | Stream baiting | Chile; 2014; TJ; Jung et al. 2017a [1] | KY788414 | KY788454 | KY788544 | KY788584 | MW427929 | KY788502 | KY788625 | MW427956 |
| **N. vietnamensis** [b] | CBS 142358; VN794 | *Castanopsis sp.,* *Acer campbellii* | Vietnam; 2016; TJ; Jung et al. 2017a [1] | KY788420 | KY788442 | KY788550 | KY788590 | MW427932 | KY788508 | KY788631 | MW427959 |
| *N. vietnamensis* [b] | CBS 142359; VN795 | *Castanopsis sp.,* *A. campbellii* | Vietnam; 2016; TJ; Jung et al. 2017a [1] | KY788421 | KY788443 | KY788551 | KY788591 | MW427931 | KY788509 | KY788632 | MW427958 |
| **N. irlandica** [bc] | CBS 147242; PR13-109 | Stream baiting | Ireland; 2015; ROH; O'Hanlon et al. 2016 [11] | MW364574 | MW364589 | MW367157 | MW367187 | MW427910 | MW367172 | MW367202 | MW427937 |
| *N. irlandica* [bc] | CBS 147243; P17-76A | Stream baiting | Ireland; 2017; ROH; t.s. | MW364571 | MW364586 | MW367154 | MW367184 | MW427907 | MW367169 | MW367199 | MW427934 |
| *N. irlandica* [bc] | -; P17-76 | Stream baiting | Ireland; 2017; ROH; t.s. | MW364570 | MW364585 | MW367153 | MW367183 | MW427906 | MW367168 | MW367198 | MW427933 |
| *N. irlandica* [bc] | -; P17-76B | Stream baiting | Ireland; 2017; ROH; t.s. | MW364572 | MW364587 | MW367155 | MW367185 | MW427908 | MW367170 | MW367200 | MW427935 |
| *N. irlandica* [bc] | -; P18-110B | Stream baiting | Ireland; 2017; ROH; t.s. | MW364573 | MW364588 | MW367156 | MW367186 | MW427909 | MW367171 | MW367201 | MW427936 |
| **N. lirii** [bc] | CBS 147293; PR12-475 | Stream baiting | Ireland; 2014; ROH; O'Hanlon et al. 2016 [12] | MW364584 | MW364599 | MW367167 | MW367197 | MW427920 | MW367182 | MW367212 | MW427947 |
| *N. lirii* [bc] | CBS 147244; P18-27B | Stream baiting | N. Ireland, UK; 2018; ROH; t.s. | MW364576 | MW364591 | MW367159 | MW367189 | MW427912 | MW367174 | MW367204 | MW427939 |
| *N. lirii* [bc] | -; P18-27A | Stream baiting | N. Ireland, UK; 2018; ROH; t.s. | MW364575 | MW364590 | MW367158 | MW367188 | MW427911 | MW367173 | MW367203 | MW427938 |

(Continued)

**Table 1.** (Continued)

| Species | Isolate numbers [a] — International and/ or local collections | Origin — Host / source | Location; year, collector; reference | GenBank accession numbers | | | | | | | |
|---|---|---|---|---|---|---|---|---|---|---|---|
| | | | | ITS | LSU | btub | hsp90 | tigA | cox1 | nadh1 | rps10 |
| *N. lirii* [bc] | –; P18-27C | Stream baiting | N. Ireland, UK; 2018; ROH; t.s. | MW364577 | MW364592 | MW367160 | MW367190 | MW427913 | MW367175 | MW367205 | MW427940 |
| *N. lirii* [bc] | –; P18-95B | Stream baiting | Ireland; 2018; ROH; t.s. | MW364578 | MW364593 | MW367161 | MW367191 | MW427914 | MW367176 | MW367206 | MW427941 |
| *N. lirii* [b] | –; P18-95C | Stream baiting | Ireland; 2018; ROH; t.s. | MW364579 | MW364594 | MW367162 | MW367192 | MW427915 | MW367177 | MW367207 | MW427942 |
| *N. lirii* [bc] | –; P18-99B | Stream baiting | Ireland; 2018; ROH; t.s. | MW364580 | MW364595 | MW367163 | MW367193 | MW427916 | MW367178 | MW367208 | MW427943 |
| *N. lirii* [bc] | –; P18-104 | Stream baiting | Ireland; 2018; ROH; t.s. | MW364581 | MW364596 | MW367164 | MW367194 | MW427917 | MW367179 | MW367209 | MW427944 |
| *N. lirii* [bc] | –; P18-105 | Stream baiting | Ireland; 2018; ROH; t.s. | MW364582 | MW364597 | MW367165 | MW367195 | MW427918 | MW367180 | MW367210 | MW427945 |
| *N. lirii* [bc] | –; P18-157b | Stream baiting | Ireland; 2018; ROH; t.s. | MW364583 | MW364598 | MW367166 | MW367196 | MW427919 | MW367181 | MW367211 | MW427946 |
| *P. rubi* [b] | CBS 967.95; ATCC 90442; IMI 355974 | *Rubus idaeus* | Scotland; 1985; JM Duncan & DM Kennedy; Robideau *et al.* 2011 | HQ643340 | HQ665306 | KU899234 | KU899391 | KX251570 | HQ708388 | KU899476 | MT198492 [d] |

[a] Abbreviations of isolates and culture collections: ATCC = American Type Culture Collection, Manassas, USA; CBS = CBS collection at the Westerdijk Fungal Biodiversity Institute (previously Centraalbureau voor Schimmelcultures), Utrecht, Netherlands; IMI = CABI Bioscience, UK; other isolate names and numbers are as given by the collectors.

[b] Isolates used in the phylogenetic analyses.

[c] Isolates used in the morphological and temperature-growth studies.

[d] Sequence retrieved from http://oomycetedb.cgrb.oregonstate.edu. MT198492 is still not released at Genbank.

To analyse the phylogenetic positions of *N. irlandica* and *N. lirii* within the genus *Nothophytophthora* a 5-partition dataset (5,492 characters) of the nuclear loci ITS, LSU, *btub*, *hsp90* and *tigA* and 3-partition dataset (1,762 characters) of the mtDNA genes *cox1*, *nadh1* and *rps10* were established. All analyses included five isolates of *N. irlandica*, 10 isolates of *N. lirii*, two isolates each of the six known *Nothophytophthora* species and *Phytophthora rubi* (CBS 967.95) as outgroup taxon. With both datasets Bayesian (BI) analyses were performed using MrBayes 3.1.2 [25,26] into partitions with the invgamma model. Four Markov chains were run for 20 M generations, sampling every 1,000 steps, and with a burn in at 8,000 trees. In addition, Maximum-Likelihood (ML) analyses were carried out using the raxmlGUI v. 2.0 [27] implementation of RAxML [28] with a GTR+G nucleotide substitution model. There were 10 runs of the ML and bootstrap ("thorough boostrap") analyses with 1,000 replicates used to test the support of the branches. Phylogenetic trees were visualized in MEGA X [29] and edited in figure editor programs. Datasets presented and trees deriving from Maximum likelihood and Bayesian analyses are available from TreeBASE (27579; http://purl.org/phylo/treebase/phylows/study/TB2: S27579).

## Morphology of asexual and sexual structures

Formation of sporangia was induced by submersing two 12–15 mm square discs cut from the growing edge of a 3–7 d old V8A colony in a 90 mm diam Petri dish in non-sterile soil extract (50 g of filtered oak forest soil in 1000 ml of distilled water, filtered after 24 h; [30]). The Petri dishes were incubated at 20°C in natural light and the soil extract was changed after 6 h [31]. Shape, type of apex, caducity and special features of sporangia and the formation of hyphal swellings were recorded after 24–48 h. For each isolate 40 sporangia and 25 zoospore cysts were measured at ×400 using a compound microscope (Zeiss Imager.Z2), a digital camera (Zeiss Axiocam ICc3) and a biometric software (Zeiss ZEN). The formation of chlamydospores and hyphal swellings was examined on V8A after 15–30 d growth at 20°C in the dark. For each isolate 40 chlamydospores and hyphal swellings chosen at random were measured under a compound microscope at ×400 [1,31].

The formation of gametangia (oogonia and antheridia) and their characteristic features were examined after 21–30 d growth at 20°C in the dark on a carrot agarose medium [32]. Isolates from both new taxa were also paired with A1 and A2 tester strains of *P. ramorum* using the method of Brasier and Kirk [33] and with A1 and A2 tester strains of *P. cinnamomi* using the method of Jung et al. [31].

## Colony morphology, growth rates and cardinal temperatures

Colony growth patterns of both *Nothophytophthora* species were described from 14–d–old cultures grown at 20°C in the dark in 90 mm plates on CA, V8A and potato dextrose agar (PDA; Oxoid Ltd., UK) [31,34,35]. For temperature-growth relationships, five and nine isolates of *N. irlandica* and *N. lirii*, respectively, were subcultured onto 90 mm V8A plates and incubated for 24 h at 20°C to stimulate onset of growth [31]. Then three replicate plates per isolate were transferred to 10, 15, 20, 25, 26, 27, 28, 29 and 30°C. Radial growth was recorded after 6 d, along two lines intersecting the centre of the inoculum at right angles and the mean growth rates (mm/d) were calculated. To determine the lethal temperature, plates showing no growth at 26, 27, 28, 29 or 30°C were re-incubated at 20°C.

## Nomenclature

The electronic version of this article in Portable Document Format (PDF) in a work with an ISSN or ISBN will represent a published work according to the International Code of

Nomenclature for algae, fungi and plants, and hence the new names contained in the electronic publication of a PLOS ONE article are effectively published under that Code from the electronic edition alone, so there is no longer any need to provide printed copies. In addition, new names contained in this work have been submitted to MycoBank from where it will be made available to the Global Names Index. The unique MycoBank number can be resolved and the associated information viewed through any standard web browser by appending the MycoBank number contained in this publication to the prefix http://www.mycobank.org/MB/. The online version of this work is archived and available from the following digital repositories: PubMed Central, LOCKSS.

## Results

### Phylogenetic analysis

Across a concatenated 7,254 character alignment of the five nuclear loci LSU, *btub*, *hsp90*, ITS and *tigA*, and the three mtDNA genes *cox1*, *nadh1* and *rps10*, *N. irlandica* had 16 unique polymorphisms and differed from *N. lirii*, *N. amphigynosa*, *N. caduca*, *N. chlamydospora*, *N. intricata*, *N. valdiviana* and *N. vietnamensis* at 96–100 (1.3–1.4%), 364–368 (5.0–5.1%), 384–394 (5.3–5.4%), 43–44 (0.6%), 231 (3.2%), 134 (1.9%) and 226 (3.1%) positions, respectively. *Nothophytophthora lirii* had 45–50 unique polymorphisms, and differed from *N. amphigynosa*, *N. caduca*, *N. chlamydospora*, *N. intricata*, *N. valdiviana* and *N. vietnamensis* at 381–387 (5.3%), 389–407 (5.4–5.6%), 104–113 (1.4–1.6%), 226–243 (3.3%), 139–148 (1.9–2.0%) and 233–240 (3.2–3.3%) positions, respectively. Apart from the partially heterozygous position 1,419 in *tigA*, all isolates of *N. irlandica* were identical across all eight loci. Conversely, within *N. lirii* the three isolates from a tributary of the Shimna River in Northern Ireland (CBS 147244, P18-27A, P18-27C) differed from the six isolates from Ireland at 31 positions. The isolates of *N. lirii* were heterozygous at 7–8, 3–4, 0–1 and 19–21 positions in *btub*, *hsp90*, ITS and *tigA*, respectively, whereas *N. irlandica* had only one heterozygous position each in ITS and *tigA*. No heterozygous positions were found in the *cox1*, *nadh1* and *rps10* sequences of any *Nothophytophthora* species. *Nothophytophthora irlandica* had in the ITS two 1bp insertions at positions 1,037 and 1,067 which were shared only with *N. chlamydospora* and *N. valdiviana* while most isolates of *N. lirii* had a unique deletion at position 427.

Since for both the nuclear 5-partition dataset and the mitochondrial 3-partition dataset the trees resulting from the BI and ML analyses had similar topologies the Bayesian trees are presented here with both Bayesian Posterior Probability values and Maximum Likelihood bootstrap values included (Figs 2 and 3; TreeBASE: 27579). In all analyses *N. irlandica*, *N. lirii* and the six known *Nothophytophthora* species formed eight distinct, strongly supported clades (Figs 2 and 3).

For the nuclear 5-partition dataset the BI analysis provided higher support for the deeper nodes than the ML analysis (Fig 2). *Nothophytophthora irlandica* and *N. lirii* were closely related and formed a fully supported clade which clustered in sister position to *N. valdiviana*. Within *N. lirii* the three isolates from a tributary of the Shimna River in Northern Ireland (CBS 147244, P18 27A, P18 27C) constituted a distinct, well supported subclade. *Nothophytophthora chlamydospora* resided in a strongly supported basal position to the *N. irlandica—N. lirii—N. valdiviana* cluster. This clade of four sterile species clustered in sister position to a clade comprising the three homothallic species *N. amphigynosa*, *N. intricata* and *N. vietnamensis*. The sterile species *N. caduca* resided in a basal position to these two clades.

The BI and ML trees of the mitochondrial 3-partition dataset had a different topology compared to the nuclear 5-loci trees and showed character conflicts at deeper nodes indicated by low support values and a polytomy (Fig 3). *Nothophytophthora irlandica* and *N. chamydospora*

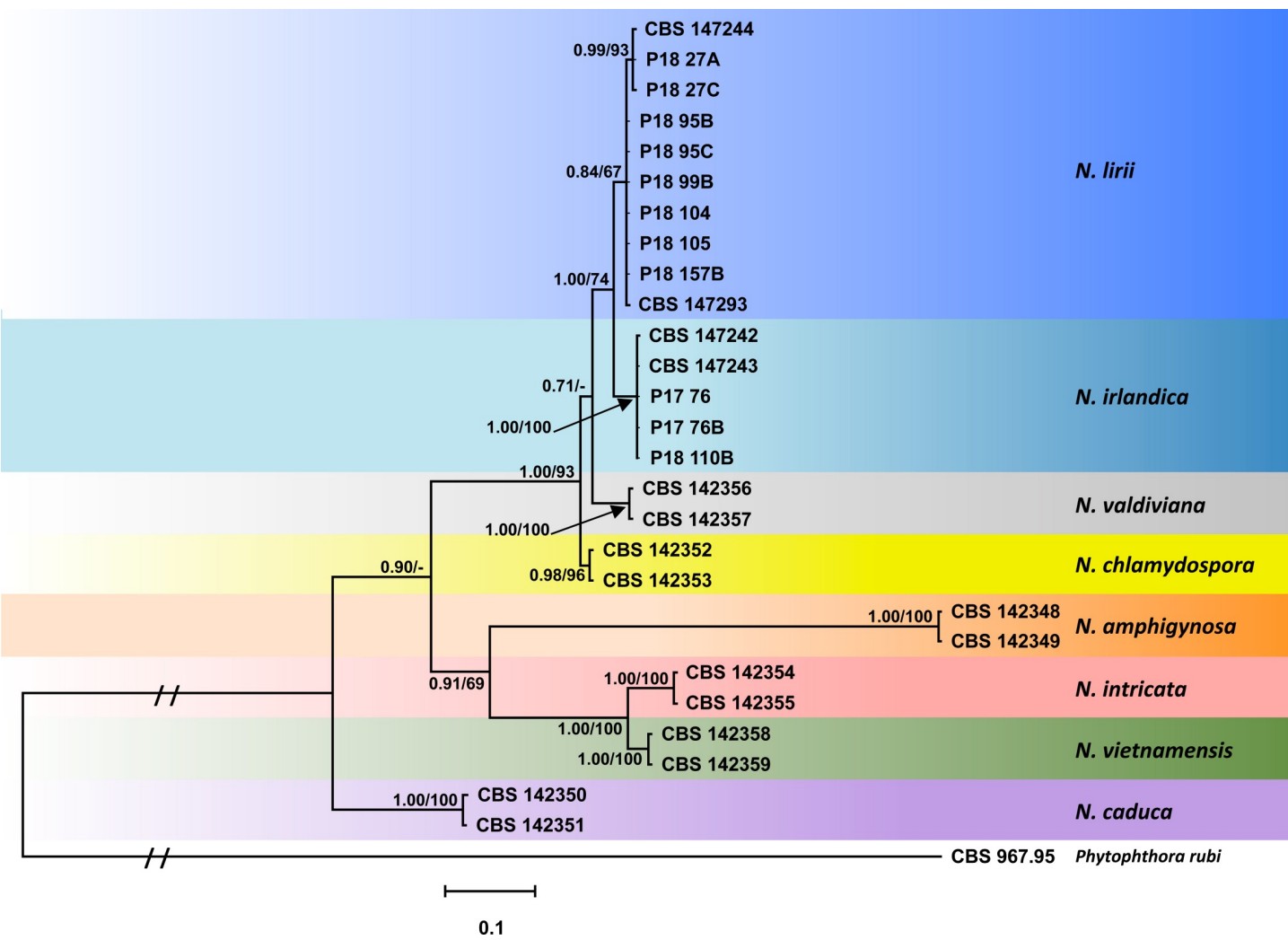

**Fig 2. Fifty percent majority rule consensus phylogram derived from Bayesian phylogenetic analysis of nuclear 5-loci (LSU, ITS, *btub*, *hsp90*, *tigA*) dataset of *Nothophytophthora irlandica* and *N. lirii* sp. nov. and six known *Nothophytophthora* species.** Bayesian posterior probabilities (left) and Maximum Likelihood bootstrap values (right; in %) are indicated, but not shown below 0.7 and 60%, respectively. *Phytophthora rubi* was used as outgroup taxon. Scale bar indicates 0.1 expected changes per site per branch.

formed a fully supported clade which resided in sister position to *N. lirii*. Similar to the nuclear analyses the three *N. lirii* isolates from a tributary of the Shimna River in Northern Ireland formed a distinct subclade separated from the Irish *N. lirii* isolates. *Nothophytophthora caduca* was basal to the *N. irlandica—N. lirii—N. chlamydospora* cluster while *N. amphigynosa* resided in sister position to *N. valdiviana* instead of clustering with the two sister species *N. intricata* and *N. vietnamensis*.

## Taxonomy

*Nothophytophthora irlandica* O'Hanlon, I. Milenković & T. Jung, (Fig 4).
 MycoBank: MB838319.
 *Etymology*: Name refers to Ireland, the region where the taxon was first found.
 *Typus*: Ireland, County Wicklow, isolated from a tributary of the Ow River in a temperate, planted coniferous forest, R. O'Hanlon, 05 December 2014 (CBS H-24576 holotype, dried

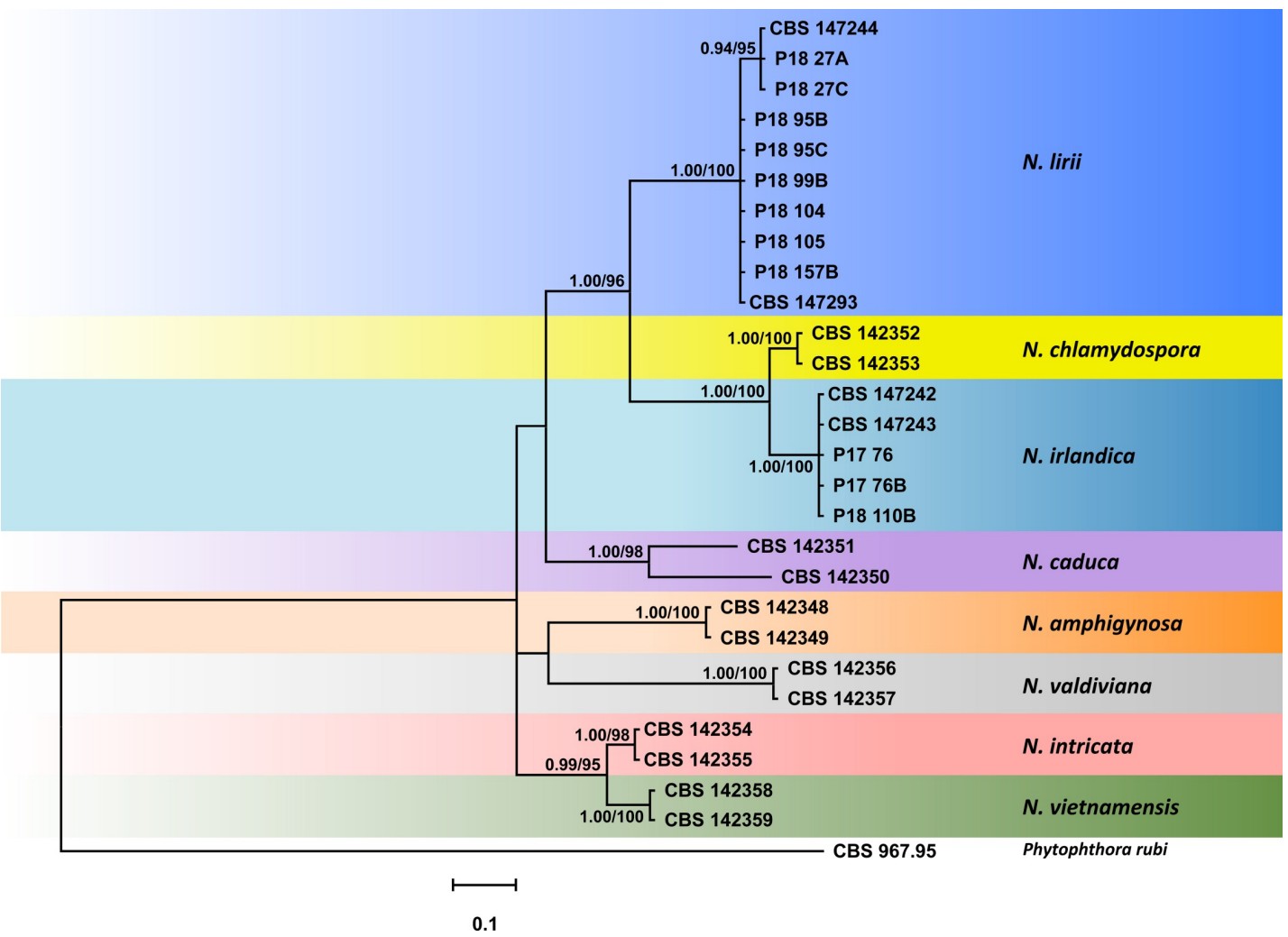

**Fig 3. Fifty percent majority rule consensus phylogram derived from Bayesian phylogenetic analysis of mitochondrial 3-loci (*cox1*, *nadh1*, *rps10*) dataset of *Nothophytophthora irlandica* and *N. lirii* sp. nov. and six known *Nothophytophthora* species.** Bayesian posterior probabilities (left) and Maximum Likelihood bootstrap values (right; in %) are indicated, but not shown below 0.7 and 60%, respectively. *Phytophthora rubi* was used as outgroup taxon. Scale bar indicates 0.1 expected changes per site per branch.

culture on CA, herbarium Westerdijk Fungal Biodiversity Institute, CBS 147242 = Pr13-109, ex-type culture). ITS and *cox1* sequences GenBank MW364574 and MW367172, respectively.

*Additional specimens*: Ireland, County Waterford. Isolated from Owenashad River in a temperate mixed coniferous and deciduous forest. Collected: R. O'Hanlon, July 2017; CBS 147243 = P17-76A, P17-76, P17-76B. July 2018; P18-110B.

Sporangia, hyphal swellings and chlamydospores (Fig 4)—Sporangia of *N. irlandica* were infrequently observed on solid V8A and were produced abundantly after 24 hr in non-sterile soil extract. Sporangia were usually borne terminally (Fig 4A–4H and 4J) or very rarely laterally on unbranched undulating sporangiophores or less frequently in dense sympodia of 2–4 sporangia (Fig 4J). Mature sporangia were non-papillate (Fig 4A–4F and 4I) and had a conspicuous opaque plug formed inside the sporangiophore close to the sporangial base which averaged 2.7 ± 0.9 μm (Fig 4A–4G and 4I). They were partially caducous breaking off just below the basal plug (Fig 4I). Sporangial shapes ranged from ovoid or elongated ovoid (28.5%; Fig 4A–4C and 4G), ellipsoid (29.3%; Fig 4E and 4I) and limoniform (41.5%; Fig 4F and 4I) to

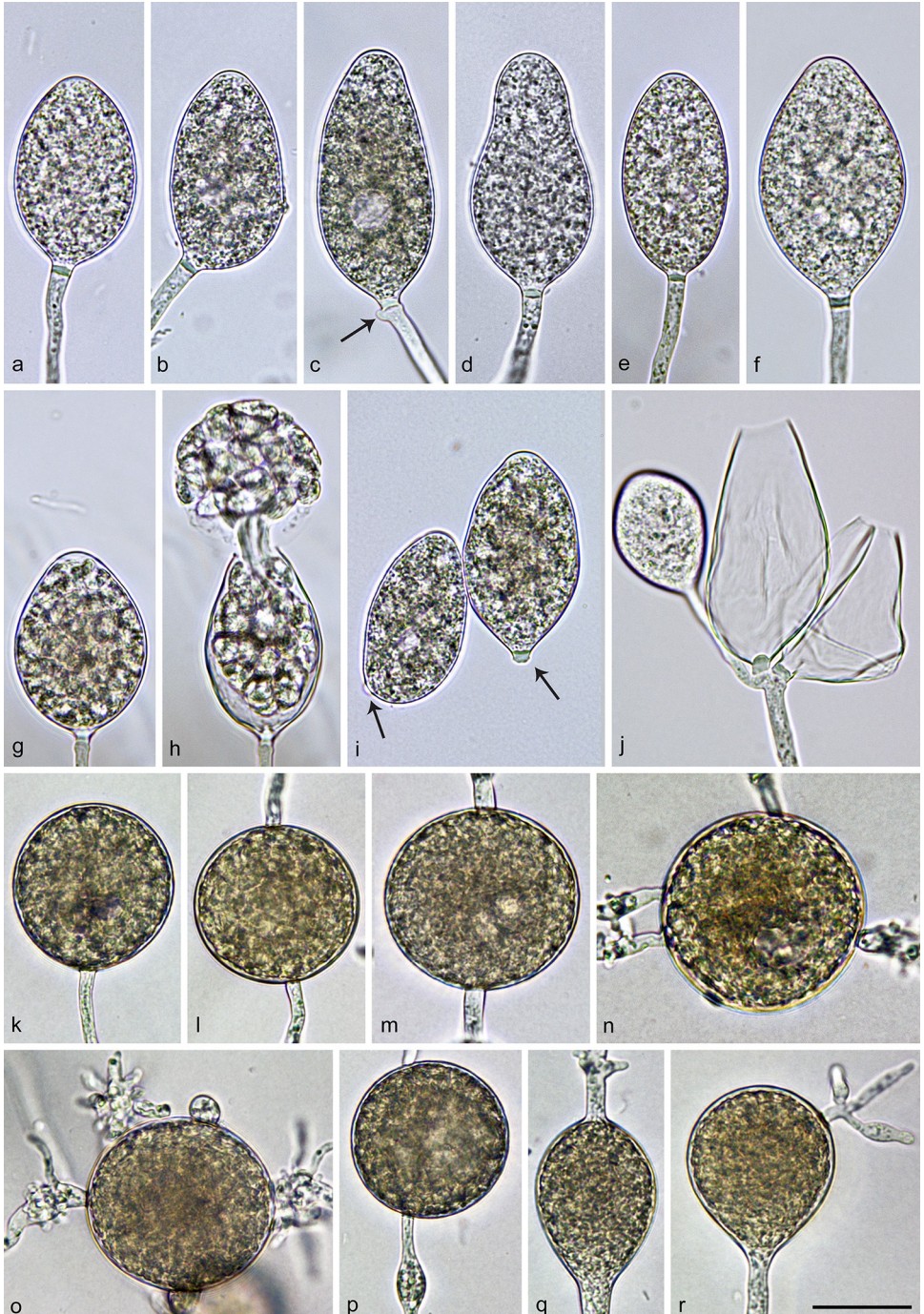

**Fig 4. Morphological structures of *Nothophytophthora irlandica*.** a–j. structures formed on V8 agar flooded with non-sterile soil extract. a–i. mature, nonpapillate, terminal sporangia with conspicuous basal plugs; a. ovoid; b. ovoid, laterally attached; c. elongated-ovoid with vacuole and beginning external proliferation (arrow); d. obpyriform; e. ellipsoid; f. limoniform; g. ovoid, before release of the fully differentiated zoospores; h. same ovoid sporangium as in g releasing zoospores; i. caducous sporangia with short pedicel–like basal plugs (arrows); j. dense sympodium with two empty sporangia after zoospore release and one immature sporangium; k–r. structures formed in solid V8 agar; k–p. globose chlamydospores; k. terminal; l–m. intercalary inserted; n–o. terminal, with radiating hyphae showing abundant production of short lateral hyphae; p. intercalary inserted with small elongated hyphal swelling; q–r. large ovoid hyphal swellings. Scale bar = 25 μm, applies to a–r.

obpyriform (<1%; Fig 4D). Sporangia with special features like lateral attachment of the sporangiophore (27.1%; Fig 4B, 4C and 4I), a vacuole (<1%; Fig 4C) or undulating sporangiophores (32.1%) occurred in all isolates. Sporangia proliferated exclusively externally, usually immediately below the sporangial base (Fig 4C and 4J). Sporangial dimensions of five isolates averaged 47.1 ± 6.1 × 28.5 ± 3.4 μm (overall range 28–74.2 × 15.9–46.6 μm and range of isolate means 44.4–51.1 × 23.3–30.7 μm). The length/breadth ratio averaged 1.7 ± 0.2 with a range of isolate means of 1.5–1.9 (Table 2). Zoospores were discharged through an exit pore 5.8–14.9 μm wide (av. 10.6 ± 1.8 μm; Fig 4H and 4J). Zoospores were limoniform to reniform whilst motile, becoming spherical (av. diam = 9.6 ± 1.3 μm) on encystment. Cysts germinated directly. Intercalary, globose to subglobose or limoniform, sometimes catenulate hyphal swellings, measuring 12.8 ± 3.8 μm, were infrequently formed on sporangiophores by all isolates. Globose (99.9%; Fig 4K–4P) or less frequently pyriform, limoniform or irregular (<1%) chlamydospores were produced terminally (Fig 4K, 4N and 4O) or intercalary (Fig 4L, 4M and 4P) and measured 42.0 ± 4.0 μm (Table 2). They often had radiating hyphae which usually showed intense and dense branching close to the chlamydospore (Fig 4N and 4O). Hyphal swellings were also observed (Fig 4Q and 4R).

Oogonia, oospores and antheridia—All five isolates of *N. irlandica* examined were self-sterile and did not form gametangia in single culture or in pairings with A1 and A2 tester strains of *P. ramorum* and *P. cinnamomi*.

Colony morphology, growth rates and cardinal temperatures (Figs 5 and 6)—Colonies of the five tested isolates of *N. irlandica* on V8A and CA were appressed to submerged and had either rosaceous or faintly striate to uniform patterns. On PDA colonies of all isolates were appressed and dense felty with a more or less clear rosaceous pattern and irregular margins (Fig 5). All five isolates of *N. irlandica* included in the temperature-growth test had similar growth rates and cardinal temperatures. The maximum and lethal growth temperatures were 25 and 30°C, respectively (Table 2, Fig 6). The average radial growth rate at the optimum temperature of 20°C was 2.1 ± 0.3 mm/d (Table 2, Fig 6).

***Nothophytophthora lirii*** O'Hanlon, I. Milenković & T. Jung, (Fig 7).

MycoBank: MB838320.

*Etymology*: Name refers to the mythological King Lir in Gaelic folklore. The Children of Lir were transformed into swans and cursed so that they could never leave certain waterbodies in Ireland. This taxon has to date only been found in waterbodies in the island of Ireland.

*Typus*: Ireland, County Waterford. Isolated from Owenashad River in a temperate mixed forest. Collected: R. O'Hanlon, 18 March 2014 (CBS H-24577 holotype, dried culture on CA, herbarium Westerdijk Fungal Biodiversity Institute, CBS 147293 = Pr12-475, ex-type culture). ITS and *cox1* sequences GenBank MW364584 and MW367182, respectively.

*Additional specimens*: UK, Northern Ireland, County Down. Isolated from a tributary of the Shimna River. Collected: R. O'Hanlon, March 2018; CBS 147244 = P18-27B, P18-27A, P18-27C. Ireland, County Waterford. Isolated from Owenashad River in a temperate mixed coniferous and deciduous forest. Collected: R. O'Hanlon, June 2018; P18-95B, P18-99B, P18-104, P18-105; August 2018; P18-157.

Sporangia, hyphal swellings and chlamydospores (Fig 7)—Sporangia of *N. lirii* were infrequently observed on solid V8A and were produced abundantly after 24 hr in non-sterile soil extract. Sporangia were borne terminally on unbranched sporangiophores (Fig 7A–7E and 7G) or less frequently laterally on short sporangiophores (Fig 7F). Sometimes secondary lateral sporangia are formed just below the empty upper section of a sporangiophore (Fig 7K) after the terminal sporangia have already released zoospores. Rarely, dense sympodia of 3 to 4 sporangia were observed (Fig 7L). Sporangia were mostly non-papillate (Fig 7A–7C and 7G) or rarely shallow semi-papillate (Fig 7D and 7E). In all mature sporangia a conspicuous opaque

**Table 2. Morphological characters and dimensions (mean ± SD; µm), cardinal temperatures (˚C) and temperature-growth relations (mm/d) on V8-juice agar[a] of *Nothophytophthora irlandica*, *N. lirii* and six known *Nothophytophthora* species (data from Jung et al. [1]).**

| | *N. irlandica* | *N. lirii* | *N. amphigynosa* | *N. caduca* | *N. chlamydospora* | *N. valdiviana* | *N. intricata* | *N. vietnamensis* |
|---|---|---|---|---|---|---|---|---|
| No. of isolates | 5 [b] | 9 [b] | 8 [b] | 14 [b] | 5 [b] | 5 [b] | 6 [b] | 8 [b] |
| Sporangia | 28.8% ovoid/ elongated ovoid, 29.6% ellipsoid, 41.7% limoniform, 1% obpyriform | 23.4% ovoid/elong. ovoid (23.4%), 31.5% ellips-oid, 40.9% limoniform, 1% obpyriform | 82% ovoid, 12% ellipsoid, 5% obpyriform (limoniform, mouse-shaped) | **83% ovoid,** 7% ellipsoid, 4% limoniform (obpyriform, pyriform, mouse-shaped) | 44% ovoid, **27.5% ellipsoid, 22.5% limoniform** (obpyriform, pyriform, mouse-shaped) | 50.5% ovoid, **40.5% limoni-form, 6% ellipsoid,** (obpy-riform, pyriform, mouse-shaped) | 71% ovoid, **15% obpyriform**, 7% limoniform, 5% ellipsoid (pyriform, mouse-shaped) | **91% ovoid,** 6% ellipsoid, 3% limoniform |
| lxb mean | **47.1 ± 6.1 × 28.5 ± 3.4** | **43.4 ± 6.5 × 25.0 ± 2.9** | **47.0±5.6 x 26.4±1.8** | 37.9±4.6 x 25.7±3.0 | 37.6±4.9 x 22.1 ±2.5 | 42.7±4.6 x 28.0±3.5 | 38.5±2.8 x 24.8±1.5 | 36.4±12.7 x 29.3±8.1 |
| range of isolate means | 44.4–51.1 × 23.3–30.7 | 36.3–46.9 × 22.6–27.8 | 41.5–52.0 x 25.4–27.3 | 34.7–43.1 x 23.3–28.2 | 35.6–38.9 x 20.4–23.2 | 40.4–44.7 x 25.6–29.5 | 37.6–40.5 x 23.4–26.3 | 34.1–37.9 x 24.1–25.8 |
| total range | 28–74.2 × 15.9–46.6 | 27.3–65.1 × 16.3–34.8 | 33.6–60.6 x 21.3–32.4 | 24.1–54.4 x 18.1–35.9 | 27.4–57.2 x 17.0–30.8 | 30.2–55.7 x 18.6–47.5 | 27.8–49.2 x 18.6–30.2 | 28.4–42.1 x 20.6–28.1 |
| l/b ratio | 1.66 ± 0.24 | 1.74 ± 0.15 | **1.78 ± 0.17** | 1.48 ± 0.15 | **1.71 ± 0.17** | 1.53 ± 0.14 | 1.55 ± 0.18 | 1.47 ± 0.08 |
| caducity | **partially caducous** | **partially caducous** | – | 32.1% (10–53%) | **25.2% (11–41%)** | 6.8% (4–10%) | – | **15.8% (4–36%)** |
| pedicel-like basal plug | 2.7 ± 0.9 | 2.7 ± 0.9 | 2.9 ± 0.6 | 2.6 ± 0.7 | 2.8 ± 1.6 | 2.4 ± 0.5 | 2.9 ± 0.7 | 2.7 ± 0.7 |
| internal proliferation | – | – | – | **nested and extended** | – | **nested and extended** | – | – |
| exitpores | 10.58 ± 1.82 | 9.3 ± 1.78 | 8.9 ± 1.4 | 10.4 ± 2.2 | 8.2 ± 1.7 | 9.4 ± 1.8 | 9.0 ± 1.6 | 7.6 ± 1.5 |
| sympodia | Infrequent, lax | infrequent, lax | **infrequent, lax** | **frequent, lax** | **frequent, lax or dense** | frequent, lax or dense | **infrequent, lax** | **frequent, lax or dense** |
| zoospore cysts | 9.64 ± 1.32 | 8.72 ± 1.63 | 9.0 ± 1.1 | 7.4 ± 0.6 | 8.6 ± 0.8 | 8.6 ± 1.1 | 8.1 ± 1.1 | 8.4 ± 0.7 |
| sporangiospore swellings | 12.8 ± 3.8; infrequent | n/a; rare | 11.1 ± 2.8; rare | 10.2 ± 2.0; rare | 15.2 ± 6.3; rare | 14.0 ± 2.7; rare | 9.8 ± 1.5; rare | n/a; rare |
| Breeding system | **self-sterile** | **self-sterile** | Homothallic | **self-sterile** | **self-sterile** | self-sterile | homothallic | homothallic |
| Oogonia | | | | | | | | |
| mean diam | – | – | 25.3 ± 1.7 | – | – | – | **30.1 ± 3.9** | **23.9 ± 3.0** |
| range of isolate means | – | – | 24.3–25.5 | – | – | – | 28.1–31.8 | 22.3–27.3 |
| total range | – | – | 18.4–29.7 | – | – | – | 16.7–41.8 | 18.6–33.0 |
| tapering base | – | – | **2.9% (0–7.5%)** | – | – | – | 7.5% (0–30%) | 75.4% (42–95%) |
| thin stalks | – | – | **58.3% (10–100%)** | – | – | – | **29.4% (2.5–45%)** | **3.1% (0–12.5%)** |
| curved base | – | – | - | – | – | – | **1.3% (0–5%)** | **24.4% (7.5–32.5%)** |
| elongated | – | – | **12.5% (5–20%)** | – | – | – | **5.6% (0–17.5%)** | **70.6% (60–85%)** |
| Oospores | – | – | | – | – | – | | |
| *plerotic oospores* | – | – | 99.2% | – | – | – | 96.9% (92.5–100%) | 96.9% (87.5–100%) |
| mean diam | – | – | 23.4 ± 1.7 | – | – | – | 28.3 ± 3.5 | 22.5 ± 2.4 |
| Total range | – | – | 17.2–28.0 | – | – | – | 15.7–38.4 | 17.6–29.5 |
| wall diam | – | – | 1.7 ± 0.3 | – | – | – | 2.1 ± 0.4 | 1.8 ± 0.3 |

*(Continued)*

**Table 2.** (Continued)

| | *N. irlandica* | *N. lirii* | *N. amphigynosa* | *N. caduca* | *N. chlamydospora* | *N. valdiviana* | *N. intricata* | *N. vietnamensis* |
|---|---|---|---|---|---|---|---|---|
| **oospore wall index** | – | – | 0.38 ± 0.05 | – | – | – | 0.38 ± 0.06 | 0.42 ± 0.05 |
| **Abortion rate** | – | – | 4.2% (1–25%) | – | – | – | 10.8% (1–18%) | 1.0% (0–4%) |
| **Antheridia** | – | – | **87.2% amphigynous** | – | – | – | **100% paragynous** | **100% paragynous** |
| **size** | – | – | 8.5±1.8 x 6.5 ±0.9 | – | – | – | 10.0±1.9 x 6.9 ±1.2 | 7.2±1.2 x 4.6 ±0.9 |
| **intricate stalks** | – | – | **28.8% (22.5–35%)** | – | – | – | **63.3% (50–72.5%)** | 46.7% (42.5–52.5%) |
| **Chlamydospores** | 99% globose, 1% pyriform; 42.0 ± 4.0 | 99% globose, 1% pyriform; 51.7 ± 6.7 | – | – | 98.1% globose, 1.9% pyriform; radiating; clusters; 43.7 ± 7.0 | – | – | – |
| **Hyphal swellings** | Globose, (limoform) 12.8 ± 3.8 | Globose, (pyriform), 14.75 ± 6 | – | – | globose, (pyri-, limoni-form); 29.2 ± 6.1 | – | – | – |
| **Lethal temperature** | 30 or 32.5 | 32.5 or 35 | 28 | 28 or 30 | 26 | 30 | 28 | 29 |
| **Maximum temperature** | 25 | 25 | 27 | 26 or 28 | 25 | 28 | 27 | 27 |
| **Optimum temperature** | 20 | 20 | 20 | 20 or 25 | 20 | 25 | 25 | 25 |
| **Growth rate at 20°C** | 2.1 ± 0.25 | 1.7 ± 0.34 | 3.1 ± 0.05 | 3.1 ± 0.21 | 3.2 ± 0.05 | 2.9 ± 0.05 | 2.2 ± 0.06 | 2.5 ± 0.04 |
| **Growth rate at 25°C** | 1.2 ± 0.18 | 1.4 ± 0.15 | 3.0 ± 0.06 | 3.6 ± 0.08 | 0.5 ± 0 | 3.1 ± 0.1 | 2.5 ± 0.07 | 2.9± 0.05 |

[a] Oogonia and oospores were studied and measured on carrot agar.

[b] Numbers of isolates included in the growth tests: *N. irlandica* = 6; *N. lirii* = 8; *N. amphigynosa* = 4; *N. caduca* = 10; *N. chlamydospora* = 4; *N. valdiviana* = 4; *N. intricata* = 5; *N. vietnamensis* = 8.

– = character not observed.

Most discriminating characters are highlighted in bold. in brackets are ranges of isolate means.

plug was formed inside the sporangiophore close to the sporangial base which averaged 2.7 ± 0.9 μm (Fig 7A–7H, 7J and 7L). Sometimes a conspicuous double plug could be observed (Fig 7E and 7G). Sporangia were partially caducous breaking off below the basal plug (Fig 7H and 7J). Sporangial shapes ranged from ovoid or elongated ovoid (23.4%; Fig 7A–7C and 7H–7J), ellipsoid or elongated ellipsoid (31.5%; Fig 7D, 7G and 7L) and limoniform (40.9%; Fig 7F and 7L) to obpyriform or elongated obpyriform (1%; Fig 7E). Sporangia with special features like lateral attachment of the sporangiophore (11.8%; Fig 7B), curved apex (1.0%; Fig 7G), a vacuole (1%; Fig 7A) or undulating sporangiophores (31.8%) occurred in all isolates. Sporangia proliferated exclusively externally, usually immediately below the old sporangium (Fig 7D, 7F, 7H and 7L). Sporangial dimensions of nine isolates averaged 43.4 ± 6.5 × 25.0 ± 2.9 μm (overall range 27.3–65.1 × 16.3–34.8 μm and range of isolate means 36.3–46.9 × 22.6–27.8 μm). The length/breadth ratio averaged 1.74 ± 0.15 with a range of isolate means of 1.6–2.0 (Table 2). In all isolates, a few sporangia failed to form a basal septum and continued to grow at the apex (Fig 7T). Zoospores were discharged through an exit pore 5.1–14.5 μm wide (av. 9.3 ± 1.8 μm; Fig 7I and 7L). Zoospores were limoniform to reniform whilst motile,

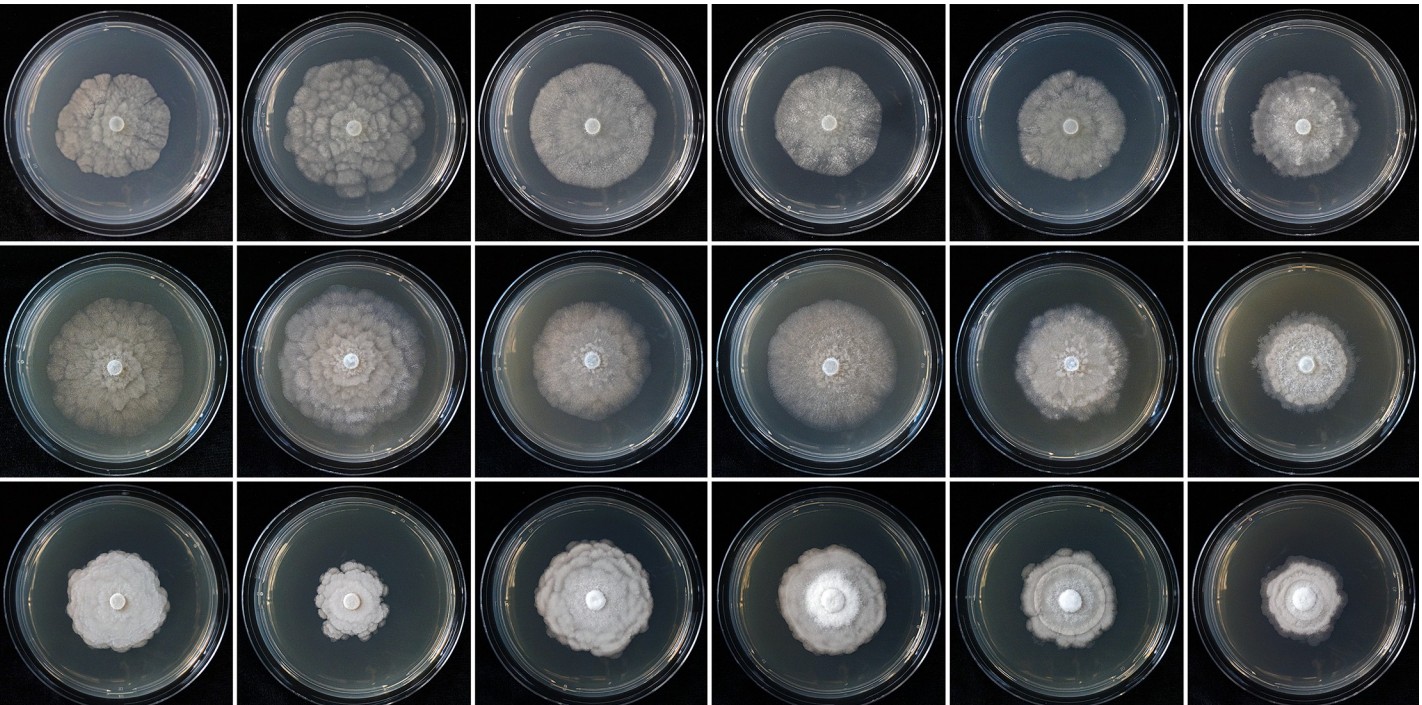

**Fig 5. Colony morphology of *Nothophytophthora irlandica* isolates CBS 147242 and P17-76, and *Nothophytophthora lirii* isolates CBS 147244, P18-105, P18-27A and P18-99B (from left to right) after 14 d growth at 20˚C on V8 agar, carrot agar and potato-dextrose agar (from top to bottom).**

becoming spherical (av. diam = 8.7 ± 1.6 μm) on encystment. Cysts germinated directly. Intercalary, globose or limoniform, sometimes catenulate hyphal swellings, measuring 14.6 ± 6 μm, were formed by all isolates. Globose (99.9%; Fig 7M–7S) or less frequently pyriform to irregular (0.1%) chlamydospores were produced terminally (Fig 7M, 7N, 7P and 7Q), laterally (Fig 7O) or intercalary (Fig 7R and 7S) and measured 51.7 ± 6.7 μm (Table 2). They sometimes had radiating irregular hyphae with small hyphal swellings (Fig 7P and 7Q). Oogonia, oospores and antheridia—all seven tested isolates of *N. lirii* were self-sterile and did not form gametangia in single culture or in pairings with A1 and A2 tester strains of *P. ramorum* or *P. cinnamomi*.

Colony morphology, growth rates and cardinal temperatures (Figs 5 and 6)—Colonies showed slight variations between the nine isolates tested. On V8A and CA they were mostly faintly radiate with limited, appressed-felty aerial mycelium in the center and often with irregular and sometimes submerged margins. On PDA colonies were dense-felty white, sometimes with faint concentric rings and always with irregular margins which were partly submerged (Fig 5). Temperature-growth relations are shown in Fig 6. All nine tested isolates had similar growth rates and cardinal temperatures. The maximum and lethal growth temperatures were 25 and 30˚C, respectively. The average radial growth rate at the optimum temperature of 20˚C was 1.7 ± 0.3 mm/d (Table 2; Fig 6).

## Notes

*Nothophytophthora irlandica* and *N. lirii* share many features with the six described *Nothophytophthora* species, including slow colony growth with relatively low maximum temperatures for growth and the production of a conspicuous opaque plug at the sporangial base. Both new *Nothophytophthora* species differ from *N. amphigynosa*, *N. caduca*, *N. intricata*, *N. valdiviana*

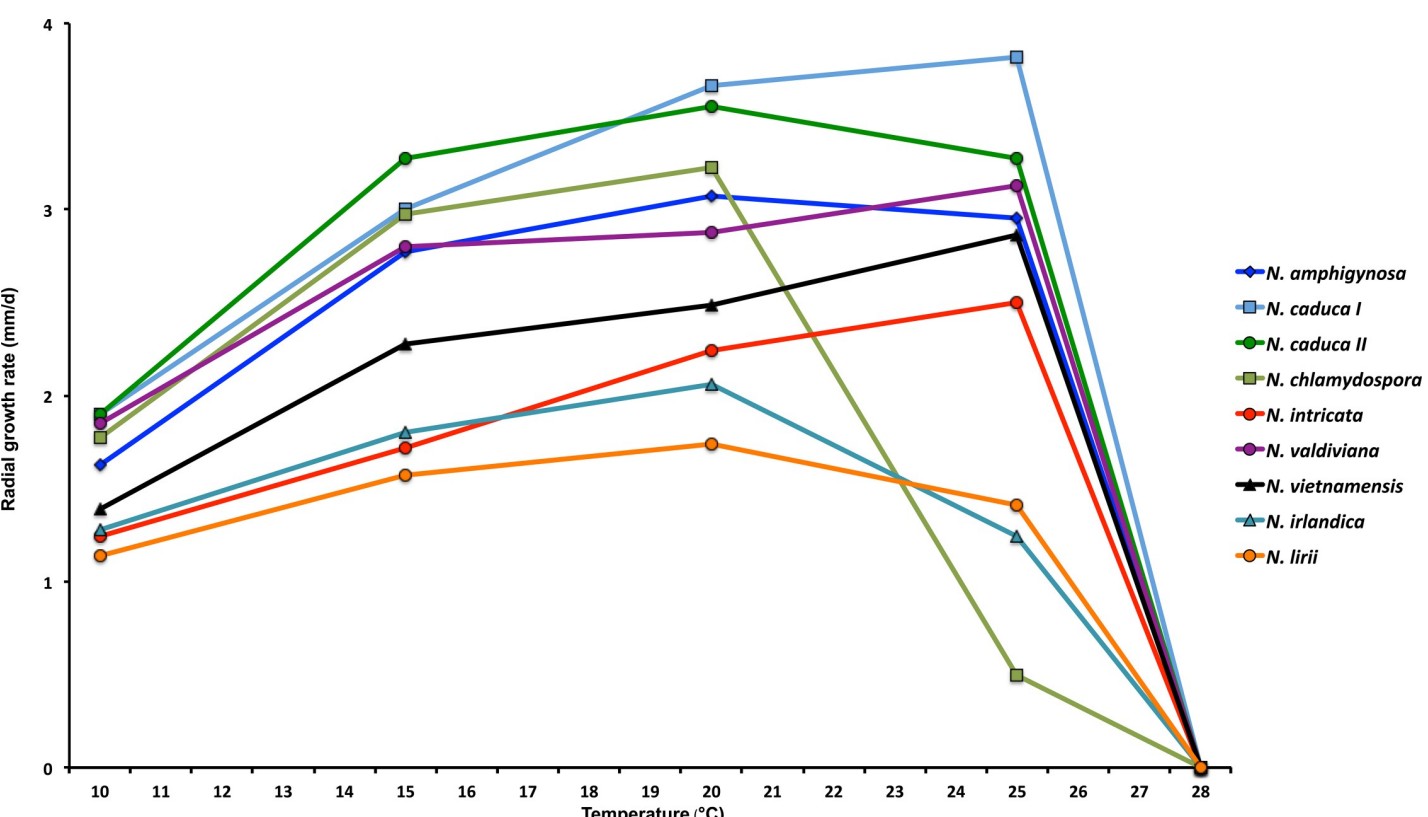

**Fig 6. Mean radial growth rates on V8 agar at different temperatures for *Nothophytophthora irlandica* (5 isolates) and *N. lirii* (9 isolates) from this study in** comparison to *N. amphigynosa*, *N. caduca*, *N. chlamydospora*, *N. intricata*, *N. valdiviana* and *N. vietnamensis* (data from Jung *et al.* 2017a [1]).

and *N. vietnamensis* by having considerably slower growth at both 20˚C and 25˚C and, in addition, from *N. intricata*, *N. valdiviana* and *N. vietnamensis* by having a lower optimum temperature for growth (20˚C vs 25˚C) (Fig 6; [1]). In addition, they are easily distinguished from *N. amphigynosa*, *N. intricata* and *N. vietnamensis* by being sterile (Table 2; [1]). *Nothophytophthora chlamydospora* is phylogenetically closest to the two new *Nothophytophthora* species and shares with them the sterile breeding system and the production of chlamydospores and of partially caducous sporangia with exclusively external proliferation (Table 2; [1]). However, *N. irlandica* and *N. lirii* can be distinguished from *N. chlamydospora* by having considerably slower growth at 15 and 20˚C and faster growth at 25˚C, by producing smaller sporangial sympodia (less than 4 sporangia vs less than 6–8 sporangia) and by the absence of secondary chlamydospores on hyphae radiating from primary chlamydospores. In addition, compared to *N. chlamydospora*, *N. lirii* and *N. irlandica* produce on average larger chlamydospores and longer sporangia, respectively. *Nothophytophthora irlandica* and *N. lirii* differ from each other in the sizes of their sporangia and chlamydospores and in their colony morphologies on V8A and CA (Table 2; Fig 5). Furthermore, *N. irlandica* and *N. lirii* formed well supported distinct clades in the BI and ML analyses of both the nuclear 5-loci and the mitochondrial 3-loci datasets.

## Hosts and geographic distribution

*Nothophytophthora irlandica* and *N. lirii* have hitherto only been detected on *R. ponticum* leaves floating naturally or as baits in streams in Ireland and Northern Ireland. Naturally fallen

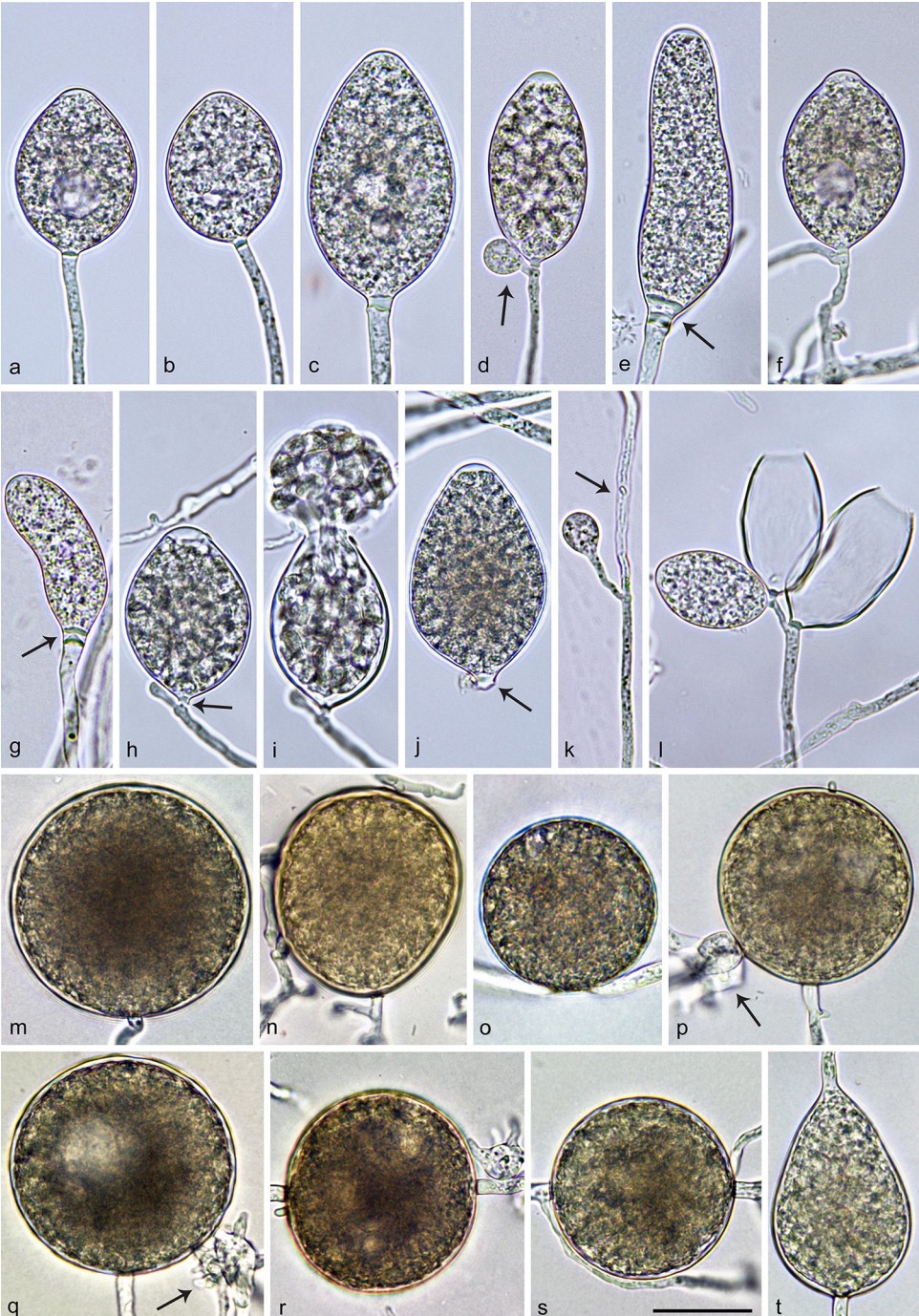

**Fig 7. Morphological structures of *Nothophytophthora lirii*.** a–l. structures formed on V8 agar flooded with non-sterile soil extract. a–j. mature sporangia with conspicuous basal plugs; a. nonpapillate, ovoid with vacuole; b. nonpapillate, ovoid, slighly laterally attached; c. nonpapillate, elongated-ovoid; d. ellipsoid, with swollen apex before zoospore release and with beginning external proliferation (arrow); e. nonpapillate, elongated-obpyriform with two basal plugs (arrow); f. nonpapillate, limoniform, on a short lateral hypha, with vacuole and external proliferation; g. nonpapillate, elongated-ellipsoid, curved, with two basal plugs (arrow); h. ovoid, with swollen apex before release of the fully differentiated zoospores, with beginning external proliferation; almost breaking-off at the basal plug (arrow); i. same ovoid sporangium as in g releasing zoospores; j. elongated-ovoid, caducous sporangium with short pedicel–like basal plug (arrow); k. secondary, lateral sporangium forming just below the empty upper section of the sporangiophore (arrow); l. dense sympodium with two empty sporangia after zoospore release and one immature limoniform sporangium; m–t. structures formed in solid V8 agar; m–s. globose or subglobose thick-walled chlamydospores; m.

terminal; n. subglobose, intercalary inserted; o. laterally sessile; p–q. terminal with a few swollen, radiating hyphae (arrows); r–s. intercalary inserted; t. obpyriform sporangium that ailed to form a basal septum and continued to grow at the apex. Scale bar = 25 μm, applies to a–t.

leaves of other tree species (e.g. *Fraxinus*, *Fagus*, *Corylus*, *Quercus*) floating in rivers at locations where the new *Nothophytophthora* species had been recovered never yielded any isolates of *Nothophytophthora*. Similarly, testing of symptomatic foliage from *R. ponticum* plants near two of these streams never yielded any isolates of *Nothophytophthora*. Several other oomycete species were recovered from the same streams, including *Phytophthora gonapodyides*, *P. chlamydospora*, *P. lacustris*, and *Elongisporangium undulatum*. Also, *P. ramorum* and *P. cactorum* were isolated from foliage of *R. ponticum* plants near the streams. Although several hundred leaves were tested for oomycetes only 15 isolates of *N. lirii* and *N. irlandica* were obtained during 2 of the 17 baiting occasions and 5 of the 15 sampling occasions of naturally fallen leaves. Therefore, neither of the two new *Nothophytophthora* species can be considered as being common in the watercourses surveyed.

## Discussion

This study has shown that the unknown oomycete isolates from streams in Ireland and Northern Ireland constitute two new distinct *Nothophytophthora* species, described here as *N. irlandica* and *N. lirii*. Both new species were differentiated from the six known *Nothophytophthora* species and from each other based on morphological characteristics, temperature-growth relationships and multi-locus phylogenetic analyses. The nuclear and mitochondrial multi-loci trees had different topologies indicating different evolutionary histories of the nuclear and mitochondrial *Nothophytophthora* genomes. Discordances between mitochondrial and nuclear genealogies are common and usually caused by incomplete lineage sorting or mitochondrial introgression [36–40]. Nonetheless, *N. irlandica*, *N. lirii* and the six known *Nothophytophthora* species formed in the BI and ML analyses of both the nuclear and mtDNA multi-locus datasets eight distinct strongly supported clades.

In the original description of the genus *Nothophytophthora* Jung et al. [1] pointed out that despite numerous oomycete surveys being carried out each year across the globe, sequences of just three strains at GenBank were matching *Nothophytophthora*. Two of these strains are designated here as ex-type isolates of *N. irlandica* (Pr13-109 = CBS 147242) and *N. lirii* (Pr12-475 = CBS 147293). A third strain, named "*Phytophthora* sp. REB326-69", was isolated from a stream in Huia in New Zealand [5] and its sequence (GenBank accession JX122744) showed 99% similarity to *N. chlamydospora* and *N. valdiviana* [1] and also to *N. irlandica* and *N. lirii*. Additional *btub* sequence screening of isolates derived from stream baiting in northern New Zealand between 2008 and 2010 [4] revealed 17 isolates in the *N. irlandica*—*N. lirii* clade (GenBank accessions MW542641–MW542657). Further characterisation of two of these isolates with *cox1* sequences (GenBank accessions MW542639 and MW542640) determined that they were *N. irlandica*. *Nothophytophthora caduca*, *N. chlamydospora* and *N. valdiviana* were described from the Valdivian region in Chile while *N. amphigynosa*, *N. intricata* and *N. vietnamensis* were first detected in Portugal, Germany and Vietnam, respectively [1]. In recent global surveys, using classical baiting tests or metabarcoding approaches, both described and unknown *Nothophytophthora* taxa were infrequently detected. These included Portugal [41], Indonesia and Japan (T. Jung, M. Horta Jung, C. M. Brasier and A. Duràn unpublished), Norway (T. Jung, T. Corcobado, I. Milenkovic and V. Talgø unpublished), Scotland [42], Czech Republic and Slovakia [7] and Spain [43]. In addition, LSU, *btub* and *cox1* sequences recently submitted to GenBank (e.g. accession nos. for isolate SM08APR_ANG1: MG685808,

MG701979, MG701951) show that *N. caduca* occurs in Californian streams, more than 10,000 km distant from the original findings in Chile [1]. Apparently, despite their occurrence in most continents, members of the genus *Nothophytophthora* are only infrequently found in oomycete surveys. The most likely explanation for the scarcity of *Nothophytophthora* records is their slow growth in culture preventing their isolation in the presence of faster growing oomycete genera, i.e. *Elongisporangium*, *Pythium*, *Phytopythium* and *Phytophthora* [1]. In the temperature-growth test of this study both *N. irlandica* and *N. lirii* showed even slower growth than the six known *Nothophytophthora* species. Thus, their consistent isolation over consecutive years from the same streams in Ireland and Northern Ireland, despite the presence of the much faster growing oomycetes *P. chlamydospora*, *P. gonapodyides*, *P. lacustris* and *E. undulatum*, indicates competitive sustainable populations.

The question arises whether the two new *Nothophytophthora* species are native or non-native to Ireland and Northern Ireland. The phylogenetic analyses of this study revealed that *N. irlandica* and *N. lirii* are closely related sister species of *N. chlamydospora* and *N. valdiviana*. Due to their close phylogenetic relatedness these four *Nothophytophthora* species must originate from the same biogeographic region, either Europe or temperate regions of South America. There are several lines of indirect evidence supporting that the species are non-native to the island of Ireland. The island of Ireland has no areas of pristine forests, with just 2% of the land area of Ireland classified as semi-natural native forests [44]. Of the total forest area of 673,000 ha, 68, 19 and 13% of the forests are composed of non-native, native or a mixture of non-native and native tree species, respectively [10]. Consequently, there are only few habitats in Ireland or Northern Ireland left undisturbed by human activities, including the inadvertent introduction of invasive plants and microorganisms to the wider environment. In recent years several *Phytophthora* species, including *P. ramorum*, *P. lateralis* and *P. kernoviae* were introduced to Irish habitats, most likely through the trade in plants-for-planting [11,45]. *Phytophthora kernoviae* has only been reported from the UK, Ireland, New Zealand and Chile [2,46–49]. *Phytophthora kernoviae* most likely originates from the Valdivian rainforests of Chile [2]. Since both *N. chlamydospora* and *N. valdiviana* also co-occur in the same forests [1,2] it seems feasible that *P. kernoviae*, *N. irlandica* and *N. lirii* were all introduced from Chile to the island of Ireland, most likely on living plants. Analogous, also the populations of *P. kernoviae* and *N. irlandica* in New Zealand might have been introduced from Chile, either directly or via the UK and Ireland as steppingstones. However, population genetic analyses of Chilean, Irish, British and New Zealand populations of *Nothophytophthora* and *P. kernoviae* are needed to confirm this hypothesis. The limited distribution of *Nothophytophthora* species in streams on the island of Ireland also points to their non-native status, with other recent surveys for *Phytophthora* in Ireland failing to isolate *Nothophytophthora* species [12,13].

Oomycetes are increasingly emerging as one of the most significant threats to global plant health [50–52]. Since all known *Nothophytophthora* isolates were recovered from waterbodies or – less frequently – rhizosphere soil, it is important to clarify whether *Nothophytophthora* species are plant pathogens or saprotrophs. Aquatic saprotrophic oomycetes, in particular *Phytophthora* species, are usually characterised by high cardinal temperatures, fast growth, a sterile breeding system, thin-walled chlamydospores, and the abundant production of non-papillate persistent sporangia with internal proliferation [53,54]. Having very slow growth, low cardinal temperatures and partially caducous sporangia with infrequent or lacking internal proliferation, *Nothophytophthora* species do not fit the profile of competitive aquatic saprotrophs [1]. Instead, a partially aerial lifestyle as leaf and shoot pathogens had been proposed with stream populations resulting at least partly from canopy drip [1]. In the natural and seminatural forests in Chile, Vietnam and Portugal from which *N. caduca*, *N. chlamydospora*, *N. valdiviana*, *N. amphigynosa* and *N. vietnamensis* were isolated, no obvious symptoms of above-ground

infections of plant tissues were noticed [1–3]. Likewise, in two of the streams where *N. irlandica* and *N. lirii* was present in Ireland and Northern Ireland, testing of attached symptomatic *R. ponticum* foliage did not reveal any *Nothophytophthora* species. Extensive ongoing tests of the potential aerial and soilborne pathogenicity and host ranges of the six known *Nothophytophthora* species, the two new *Nothophytophthora* species from Ireland and other yet undescribed *Nothophytophthora* species are currently being performed and their results will help to understand the lifestyle and pathological importance of *Nothophytophthora* species. Given that both of the species described here produce chlamydospores abundantly, and these structures are known to aid in survival of biologically unfavourable periods and in long-distance spread, the risk of these species spreading in plant trade should be assessed [55,56].

## Acknowledgments

SEB acknowledges P.J. Lockhart (Massey University) for review of the manuscript. TJ, MHJ, IM, MT, JJ and TK acknowledge Aneta Bačová, Henrieta Ďatková and Milica Raco (all Mendel University in Brno) for much appreciated technical support.

## Author Contributions

**Conceptualization:** Richard O'Hanlon, Marilia Horta Jung, Thomas Jung.

**Formal analysis:** Richard O'Hanlon.

**Funding acquisition:** Richard O'Hanlon, Thomas Jung.

**Investigation:** Richard O'Hanlon, Maria Destefanis, Ivan Milenković, Stanley E. Bellgard, Bevan S. Weir, Marilia Horta Jung, Thomas Jung.

**Methodology:** Richard O'Hanlon, Ivan Milenković, Michal Tomšovský, Josef Janoušek, Tomáš Kudláček, Marilia Horta Jung, Thomas Jung.

**Resources:** Richard O'Hanlon, Maria Destefanis, Thomas Jung.

**Writing – original draft:** Richard O'Hanlon, Thomas Jung.

**Writing – review & editing:** Richard O'Hanlon, Thomas Jung.

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
