## [Decision Letter · Decision Letter 0]

8 Mar 2021

PONE-D-21-03306

Two new Nothophytophthora species from streams in Ireland and Northern Ireland: Nothophytophthora irlandica and N. lirii sp. nov.

PLOS ONE

Dear Dr. O'Hanlon,

Thank you for submitting your manuscript to PLOS ONE. After careful consideration, we feel that it has merit but does not fully meet PLOS ONE’s publication criteria as it currently stands. Therefore, we invite you to submit a revised version of the manuscript that addresses the points raised during the review process.

While both reviewers agreed that the manuscript is well written and adds to our knowledge of an important new genus (Nothophytophthora), Reviewer #1 does express some valid concerns about the framing of the discussion. The manuscript could therefore be improved with any additional data you may have on pathogenicity/virulence of your new isolates, host range, and frequency of isolation. The addition of these supporting details, and appropriate text in the discussion, will increase the impact of your careful and otherwise thorough work on these two new species. Thank you for addressing these concerns in a revised manuscript.

We look forward to receiving your revised manuscript.

Kind regards,

Jaime E. Blair, PhD

Academic Editor

PLOS ONE

Journal Requirements:

2. Please include your tables as part of your main manuscript and remove the individual files. Please note that supplementary tables (should remain/ be uploaded) as separate "supporting information" files

3. Thank you for submitting the above manuscript to PLOS ONE. During our internal evaluation of the manuscript, we found significant text overlap between your submission and the following previously published works.

- https://doi.org/10.3767/persoonia.2017.39.07

We would like to make you aware that copying extracts from previous publications, especially outside the methods section, word-for-word is unacceptable, even for works which you authored. In addition, the reproduction of text from published reports has implications for the copyright that may apply to the publications.

Please revise the manuscript to rephrase the duplicated text, cite your sources, and provide details as to how the current manuscript advances on previous work. Please note that further consideration is dependent on the submission of a manuscript that addresses these concerns about the overlap in text with published work.

Additional Editor Comments:

Thank you for your submission. Please carefully address the comments of Reviewer #1 in your revision.

Reviewers' comments:

Reviewer's Responses to Questions

**Comments to the Author**

1. Is the manuscript technically sound, and do the data support the conclusions?

Reviewer #1: Partly

Reviewer #2: Yes

2. Has the statistical analysis been performed appropriately and rigorously? 

Reviewer #1: N/A

Reviewer #2: Yes

3. Have the authors made all data underlying the findings in their manuscript fully available?

Reviewer #1: Yes

Reviewer #2: Yes

4. Is the manuscript presented in an intelligible fashion and written in standard English?

Reviewer #1: Yes

Reviewer #2: Yes

5. Review Comments to the Author

Reviewer #1: This is a well-performed study and a well-written and rigorous manuscript that reports on the isolation and identification of two new Nothophytophthora spp. in Ireland, including their formal description. Generally, there are no major issues and the species descriptions are complete, well-illustrated and thoroughly performed.

The only point that leaves a feeling of disappointment to the reviewer is the fact that the question of pathogenicity of the genus/ the new species does not receive the amount of (practical) attention that it would deserve, in particular when comparing with the huge number of details given for the phylogenetic analyses, the tree topologies and the morphological description. Line 411 states that “the question arises whether the two new Nothophytophthora species are native or introduced…”. While academically, such question could of course be of interest for any microbe found anywhere in the world, in the present context it is clear that this question is of interest only in a phytopathological sense. Much of the last two pages of the discussion refers to questions of invasive microorganisms and their potential threat to plant health. Given all this, it is hard to understand why not a single, simple, and basic test of pathogenicity has been performed during the study, e.g. a leaf inoculation test with Rhododendron, instead of manoeuvring around this issue on two pages of discussion. Compared to the analysis of 8 genes, state-of-the-art phylogenetics, and the detailed description of every single aberrant form of sporangial shape, this is really a bit disappointing – while on the other hand the authors’ effort and the accurateness of the study are of course highly appreciated.

Similar to that, the point “hosts and distribution” is also only briefly covered, and the reader only gets very short information on isolation success (i.e. were the new species regularly found, or only rarely?, how many leaves were actually infected as compared to the Phytophthora spp.?, how many leaves were analysed?…). I feel that this would also contribute to the full picture that the reader should get of the new species.

Also, the info in line 448 ff. could have been described somewhat more prominently (i.e., in the results). The fact that Rhododendron leaves were not infected is explained here by a proposed equilibrium of host and pathogen, as proposed for P. ramorum in Vietnam (line 453). In this case, however, this contradicts what is stated in lines 416 ff (origin in South America). In particular, the association with Rhododendron would be of such great interest, as this genus obviously grows wild in Ireland, and was an effective bait plant! – again a case for a pathogenicity test!

Minor/ other issues:

- line 20/21: something went wrong with this sentence

- line 107: why two different primer pairs for cox?

- line 243 and elsewhere (see below): please check Figure legends, 4i does not show sympodia

- line 255: two decimal places suggest a degree of accuracy that does not exist.

- line 257 ff/ 317 ff: has this result been repeated? Might this be due to some unknown external factor? Is this also observed in other species? What might be the background/ reason? What might be the effect? Should this be mentioned in the discussion (or otherwise just left out completely)?

- line 266: there is no Fig. 4s

- line 266/267: this is purely hypothetical. Here, a chlamydospore “becomes” a hyphal swelling, while there seems to be no big difference to a sporangium becoming a hyphal swelling in line 315. This is not convincing

- line 296: Fig. 7l is not unbranched

- line 297: sporangiophores

- line 315: see above, purely hypothetical

- line 321: structure in Fig. 7t is much larger and defined differently above (in line 315)

- line 384: Than et al. to be replaced by [5]

- line 427: Reference to be replaced by [number]

Reviewer #2: Dear Dr. O'Hanlon

Corresponding author

PONE-D-21-03306, PLOS ONE

I hope this message finds you well. It was my great honor to review the manuscript ID: PONE-D-21-03306, entitled "Two new Nothophytophthora species from streams in Ireland and Northern Ireland: Nothophytophthora irlandica and N. lirii sp. nov.". I really enjoyed reading it describing two new Nothophytophthora species. The manuscript has written well and I believe it really falls within the scope of PLOS ONE. Therefore, my recommendation would be "Accept" without any revision.

Best regards,

Sonia

6. PLOS authors have the option to publish the peer review history of their article (what does this mean?). If published, this will include your full peer review and any attached files.

Reviewer #1: No

Reviewer #2: **Yes: **SONIA AGHIGHI

---

## [Author Response · Author response to Decision Letter 0]

22 Mar 2021

Dear editor

Thank you and the two expert reviewers for their constructive comments on our manuscript. Below we have laid out the reviewer, and editor comments, and suitable responses or rebuttals. References to line numbers refer to the CLEAN version of the manuscript. We have also included higher quality TIFF versions of Figs. 1, 4, 5, 7. We have included a higher quality PNG version of Fig. 6. 

Please do not hesitate to contact me if you need further information or clarification. 

Regards

Richard O’Hanlon

Editor 1: While both reviewers agreed that the manuscript is well written and adds to our knowledge of an important new genus (Nothophytophthora), Reviewer #1 does express some valid concerns about the framing of the discussion. The manuscript could therefore be improved with any additional data you may have on pathogenicity/virulence of your new isolates, host range, and frequency of isolation. The addition of these supporting details, and appropriate text in the discussion, will increase the impact of your careful and otherwise thorough work on these two new species.

Response: Thank you for your comments. We have revised the manuscript and corrected most of the points raised by the reviewers. One point which we feel we can address by way of changes to the text is the reviewer 1 query about pathogenicity tests. See more detailed responses below. 

E2: Please include your tables as part of your main manuscript and remove the individual files. Please note that supplementary tables (should remain/ be uploaded) as separate "supporting information" files

Response: Done, tables 1 and 2 added to the main text, at the end

E3: Thank you for submitting the above manuscript to PLOS ONE. During our internal evaluation of the manuscript, we found significant text overlap between your submission and the following previously published works https://doi.org/10.3767/persoonia.2017.39.07

Response: The significant text overlap detected is due to similarities in the methods section, and perhaps details of Table 1 of this manuscript, and the article of Jung et al. (2017) describing the new genus Nothophytophthora. The morphological, physiological, sequencing and phylogenetic studies of this work were carried out in the same lab and partly by the same scientists as the work for the original description of the genus Nothophytophthora (Jung et al. 2017). We used very similar methods and references, and for comparisons between the 2 new and the 6 known Nothophytophthora species we included details of many isolates of the 6 known Nothophytophthora species as Jung et al. (2017) in this manuscript. Therefore it is reasonable that much of the relevant text is quite similar. In all cases, where needed, we have made references to the Jung et al. (2017) article. 

E4: Please take this opportunity to be sure you have met all of our guidelines for new species. When publishing papers that describe a new fungal taxon name, PLOS aims to comply with the requirements of the International Code of Nomenclature for algae, fungi, and plants (ICN). The following guidelines for publication in an online-only journal have been agreed such that any scientific fungal name published by us is considered effectively published under the rules of the Code.

Response: We have considered the detailed feedback from PLOS ONE. Several co-authors of this manuscript have published during their careers valid descriptions of more than 40 new oomycete species and can confirm that we have included all relevant information and text to fulfil the requirements for the descriptions of new species of fungi/oomycetes according to the rules of the International Code of Nomenclature for algae, fungi, and plants (ICN). The name and details of a new fungal/oomycete species have to be registered either at MycoBank or Index Fungorum. We prefer to use MycoBank as it is more commonly used in mycology and especially with oomycete descriptions. The mycobank registrations numbers are given (LN 239, 284)

Reviewer 1.1 This is a well-performed study and a well-written and rigorous manuscript that reports on the isolation and identification of two new Nothophytophthora spp. in Ireland, including their formal description. Generally, there are no major issues and the species descriptions are complete, well-illustrated and thoroughly performed.

Response: thank you for this kind comment. 

R1.2 The only point that leaves a feeling of disappointment to the reviewer is the fact that the question of pathogenicity of the genus/ the new species does not receive the amount of (practical) attention that it would deserve, in particular when comparing with the huge number of details given for the phylogenetic analyses, the tree topologies and the morphological description. Line 411 states that “the question arises whether the two new Nothophytophthora species are native or introduced…”. While academically, such question could of course be of interest for any microbe found anywhere in the world, in the present context it is clear that this question is of interest only in a phytopathological sense. Much of the last two pages of the discussion refers to questions of invasive microorganisms and their potential threat to plant health. Given all this, it is hard to understand why not a single, simple, and basic test of pathogenicity has been performed during the study, e.g. a leaf inoculation test with Rhododendron, instead of manoeuvring around this issue on two pages of discussion. Compared to the analysis of 8 genes, state-of-the-art phylogenetics, and the detailed description of every single aberrant form of sporangial shape, this is really a bit disappointing – while on the other hand the authors’ effort and the accurateness of the study are of course highly appreciated.

Response: We thank the reviewer for this point. While we agree that the search for the native range does have important pathogenicity implications, we also feel that finding the native range of such microorganisms is also inherently interesting, and interesting as it can highlight routes by which invasive microorganisms are generally spreading. 

We agree with the reviewer that pathogenicity tests would be worthwhile, however we feel they are outside of the scope of this purely mycological paper. At Mendel University in Brno, one of the main topics of a currently ongoing PhD study is the extensive testing of the potential aerial and soilborne pathogenicity and host ranges of the 6 known Nothophytophthora species, the 2 new Nothophytophthora species from Ireland and another 4 yet undescribed Nothophytophthora species from Europe and Japan. We do not want to anticipate and, hence, partly spoil this very detailed and thorough phytopathological study by performing and publishing in the frame of the present study and manuscript a small-scale preliminary path trial with just the 2 new Nothophytophthora species and Rhododendron leaves. However, we added a sentence in the Discussion (LN 457-461) regarding these ongoing extensive pathogenicity tests. In order to remove some of the emphasis that the reviewer feels is placed on plant pathology and pathogenicity, we have removed text from the discussion. We have also tweaked the discussion to examine native vs non-native, instead of native vs introduced. This hopefully fits more with a narrative of the ecology of the microorganisms, rather than a purely phytopathological view that the reviewer commented upon (e.g. LN 416-417). We feel the manuscript is now more in line with other new species descriptions in mycology, and hope the reviewer agrees. We have also provided additional data to show that our field collections do not indicate strong pathogenic abilities (LN 364-365, 438-440; 454-456). 

R1.3 Similar to that, the point “hosts and distribution” is also only briefly covered, and the reader only gets very short information on isolation success (i.e. were the new species regularly found, or only rarely?, how many leaves were actually infected as compared to the Phytophthora spp.?, how many leaves were analysed?…). I feel that this would also contribute to the full picture that the reader should get of the new species

Response: we have added text along the lines of what the reviewer has requested. The additional text in the methods section (LN 77-92), to explain the sampling occasions, and in the results section (LN361-372) to show the relative frequency of the two new Nothophytophthora species .We feel this is an important addition, as suggested by the reviewer, since it illustrates the point that even in these locations where we isolated the two new Nothophytophthora species, they were in no way common and were actually only isolated 15 times from many hundreds of leaves tested. We have also added detail to the discussion to indicate this (LN438-440, 454-456).

R1.4 Also, the info in line 448 ff. could have been described somewhat more prominently (i.e., in the results). The fact that Rhododendron leaves were not infected is explained here by a proposed equilibrium of host and pathogen, as proposed for P. ramorum in Vietnam (line 453). In this case, however, this contradicts what is stated in lines 416 ff (origin in South America). In particular, the association with Rhododendron would be of such great interest, as this genus obviously grows wild in Ireland, and was an effective bait plant! – again a case for a pathogenicity test!

Response: we have added text to the results (LN361-372) and discussion (LN 454-456) to indicate that the species were not found infecting nearby rhododendron. We have also deleted the text referring to Vietnam, as the reviewer was right and it does not fit with the narrative of the species having limited pathogenic ability. 

R1.5 line 20/21: something went wrong with this sentence

Response: we have removed the duplicated text. Thank you for picking this up.

R1.6 line 107: why two different primer pairs for cox?

Response: It is a problem for cox1 sequence analyses and for phylogenetic analyses that some researchers use for cox1 primer pair COXF4N / COXR4N whereas others use primer pair FM84 / FM85 because the regions sequenced with these primer pairs have only a relatively small overlap. We always sequence with both primer pairs to get a longer sequence of this highly variable, important gene region and to allow comparisons with all other cox1 sequences submitted to GenBank.

R1.7 line 243 and elsewhere (see below): please check Figure legends, 4i does not show sympodia

Response: thanks for spotting this mistake; we have changed the reference to the correct Fig. 4j.

R1.8 line 255: two decimal places suggest a degree of accuracy that does not exist.

Response: we have changed numbers throughout the text to 1 decimal place. 

R1.9 line 257 ff/ 317 ff: has this result been repeated? Might this be due to some unknown external factor? Is this also observed in other species? What might be the background/ reason? What might be the effect? Should this be mentioned in the discussion (or otherwise just left out completely)?

Response: we have reconsidered this point, and removed the two points about these abnormal zoospores as we do not have enough information to say whether they are a useful diagnostic characteristic. At the moment, we also don' t know why the zoospores are doing this but we observed this unusual characteristic also in several Phytophthora species and are studying what might be the evolutionary significance if it has one.

R1.10 line 266: there is no Fig. 4s

Response: changed to Fig 4q, r

R1.11 line 266/267: this is purely hypothetical. Here, a chlamydospore “becomes” a hyphal swelling, while there seems to be no big difference to a sporangium becoming a hyphal swelling in line 315. This is not convincing

Response: Thank you for this comment. We have changed this to “hyphal swellings were also observed” (LN271)

R1.12 line 296: Fig. 7l is not unbranched

Response: the reviewer is right. We have changed this to “Fig. 7a-e, g” 

R1.13 line 297: sporangiophores

Response: fixed

R1.14 line 315: see above, purely hypothetical

Response: The structure is clearly a sporangium that failed to produce a basal septum and continued to grow at the apex. Functionally this structure is now just a swelling of the hypha. However, we removed " functionally becoming hyphal swellings". We also changed the legend to Figure 7t accordingly to " obpyriform sporangium that ailed to form a basal septum and continued to grow at the apex". (LN 690)

R1.15 line 321: structure in Fig. 7t is much larger and defined differently above (in line 315)

Response: Thank you for this comment. The structure is clearly a sporangium that failed to produce a basal septum and continued to grow at the apex. We changed the legend to Figure 7t accordingly to "obpyriform sporangium that ailed to form a basal septum and continued to grow at the apex". We have also removed the in text reference to Fig 7t for the sentence the reviewer pointed out. 

R1.16 line 384: Than et al. to be replaced by [5]

Response: done

R1.17 line 427: Reference to be replaced by [number]

Response: done

R2.1: I hope this message finds you well. It was my great honor to review the manuscript ID: PONE-D-21-03306, entitled "Two new Nothophytophthora species from streams in Ireland and Northern Ireland: Nothophytophthora irlandica and N. lirii sp. nov.". I really enjoyed reading it describing two new Nothophytophthora species. The manuscript has written well and I believe it really falls within the scope of PLOS ONE. Therefore, my recommendation would be "Accept" without any revision.

Response: Thank you very much for this kind comment.

---

## [Decision Letter · Decision Letter 1]

8 Apr 2021

Two new Nothophytophthora species from streams in Ireland and Northern Ireland: Nothophytophthora irlandica and N. lirii sp. nov.

PONE-D-21-03306R1

Dear Dr. O'Hanlon,

We’re pleased to inform you that your manuscript has been judged scientifically suitable for publication and will be formally accepted for publication once it meets all outstanding technical requirements.

Kind regards,

Jaime E. Blair, PhD

Academic Editor

PLOS ONE

Additional Editor Comments (optional):

Thank you for your very careful revision and for addressing the reviewer comments. Your manuscript is now acceptable for publication.

Reviewers' comments:

Reviewer's Responses to Questions

**Comments to the Author**

1. If the authors have adequately addressed your comments raised in a previous round of review and you feel that this manuscript is now acceptable for publication, you may indicate that here to bypass the “Comments to the Author” section, enter your conflict of interest statement in the “Confidential to Editor” section, and submit your "Accept" recommendation.

Reviewer #1: All comments have been addressed

2. Is the manuscript technically sound, and do the data support the conclusions?

Reviewer #1: (No Response)

3. Has the statistical analysis been performed appropriately and rigorously? 

Reviewer #1: (No Response)

4. Have the authors made all data underlying the findings in their manuscript fully available?

Reviewer #1: (No Response)

5. Is the manuscript presented in an intelligible fashion and written in standard English?

Reviewer #1: (No Response)

6. Review Comments to the Author

Reviewer #1: (No Response)

7. PLOS authors have the option to publish the peer review history of their article (what does this mean?). If published, this will include your full peer review and any attached files.

Reviewer #1: No

---

## [Editor Report · Acceptance letter]

5 May 2021

PONE-D-21-03306R1 

Two new *Nothophytophthora* species from streams in Ireland and Northern Ireland: *Nothophytophthora irlandica and N. lirii* sp. nov. 

Dear Dr. O'Hanlon:

I'm pleased to inform you that your manuscript has been deemed suitable for publication in PLOS ONE. Congratulations! Your manuscript is now with our production department. 

Kind regards, 

on behalf of

Dr. Jaime E. Blair 

Academic Editor

PLOS ONE